# Local Success Does Not Compose: Benchmarking Large Language Models for Compositional Formal Verification

**Xu Xu**[1*]   **Xin Li**[2*]   **Xingwei Qu**[3]
**Jie Fu**[4]   **Binhang Yuan**[1†]

[1]The Hong Kong University of Science and Technology
[2]Nanyang Technological University
[3]University of Manchester
[4]Shanghai Artificial Intelligence Laboratory

## Abstract

Despite rapid advances in code generation, current Large Language Models (LLMs) still struggle with reliable and verifiable code synthesis in the presence of compositional reasoning requirements across multi-function programs. To study this systematically, we introduce DafnyComp, a benchmark for generating compositional Dafny specifications for programs consisting of multiple interacting functions with non-trivial data dependencies. Unlike prior benchmarks that focus primarily on single-function annotation, DafnyComp targets programs composed of 2–5 functions arranged in acyclic call graphs, requiring specifications that establish correctness across component boundaries. DafnyComp contains 400 automatically synthesized programs: 300 chain-structured instances and 100 non-chain DAG instances generated from 10 topology templates. We evaluate frontier LLMs from major providers under a unified prompting and verification protocol. While these models achieve high syntactic well-formedness ($>99\%$) and moderate end-to-end verification ($>58\%$) on prior single-function Dafny verification benchmarks, they obtain near-zero end-to-end verification on DafnyComp. On the chain split, even the strongest evaluated model reaches only 2% verification at Pass@8, with most models below 1%; the difficulty persists under broader topologies and stronger test-time scaling. Our analysis identifies three recurring failure modes that hinder cross-functional reasoning: *specification fragility*, *implementation–proof* misalignment, and *reasoning instability*. DafnyComp provides a diagnostic benchmark for tracking progress in verifiable code generation, highlighting that bridging local correctness to compositional verification remains a key open challenge.

## 1 Introduction

Large language models (LLMs) have transformed software development through their remarkable code generation capabilities, enabling developers to produce complex programs from natural language descriptions (Chen et al., 2021; Austin et al., 2021). These advances have driven widespread adoption of programming assistants and development environments, fundamentally transforming how modern software is developed. As LLM-generated code becomes increasingly integrated into production systems, a critical question emerges: *how to ensure the correctness of automatically synthesized programs*. Unlike human-written code that can be manually reviewed and tested, the scale and complexity of LLM outputs demand systematic approaches to verification that go beyond traditional debugging methods. On the other hand, conventional testing provides only partial confidence and cannot rule out rare corner cases or subtle specification mismatches.

Formal verification provides a principled solution to this challenge by offering mathematical guarantees of program correctness through rigorous specification and proof techniques. Programming languages like Dafny enable developers to express precise contracts—preconditions, postconditions,

---

*Equal contribution.    †Corresponding author (`biyuan@ust.hk`).

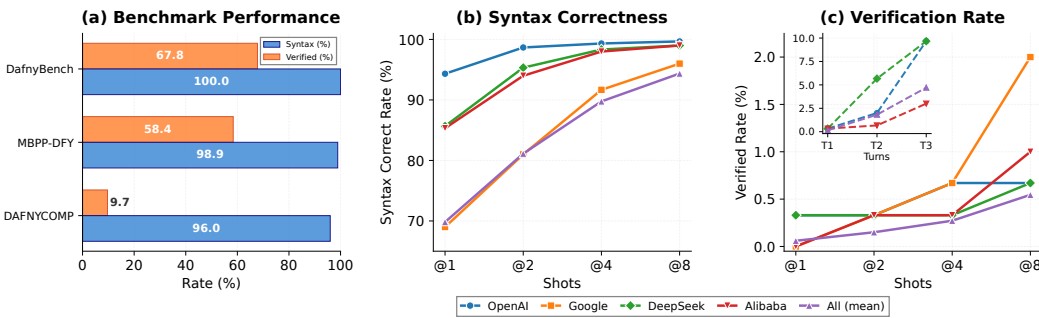

Figure 1: **The formal verification gap: high syntax success versus low verified rates.** (a) DAFNY-COMP exhibits a large Syntax–Verified gap. (b–c) Syntax improves quickly with sampling, while verification remains low; inset: multi-turn refinement helps but does not saturate.

and invariants—that can be mechanically verified against implementations (Nipkow et al., 2012). However, the adoption of formal verification has historically been constrained by what we often refer to as the "specification bottleneck": writing comprehensive annotations not only demands specialized expertise but also produces specification code that is comparable in size to the implementation itself (Cavada et al., 2025; Poesia et al., 2024). Recent research has explored the use of LLMs to automate this specification generation process, demonstrating promising results where models can complete missing annotations for individual functions and achieve moderate verification success rates (Loughridge et al., 2024; Yan et al., 2025). However, current work in this area often suffers from a critical limitation: existing benchmarks, such as DAFNYBENCH, primarily evaluate annotation completion within isolated functions (Loughridge et al., 2024), failing to address the compositional reasoning ability required for real-world, sophisticated software systems, where correctness emerges from complex interactions between multiple components (Keysers et al., 2019).

To fill this gap, we introduce DAFNYCOMP, the first benchmark explicitly designed to evaluate the generation of compositional specifications for programming languages equipped with formal verification. Concretely, we make the following contributions:

**Contribution 1.** To address the limitations of prior verification benchmarks, we present DAFNY-COMP, a new benchmark explicitly designed for compositional formal verification (§3). Unlike existing datasets such as DAFNYBENCH (Loughridge et al., 2024) that focus on specification generation for isolated single functions, DAFNYCOMP evaluates LLMs on programs composed of multiple interacting functions with real data dependencies. The benchmark consists of 400 Dafny programs synthesized by combining 2–5 independent functions into acyclic call graphs, including 300 chain-structured programs and a 100-program non-chain DAG extension, forcing models to reason across function boundaries to ensure end-to-end correctness. Note that our design is the first to require actual compositional reasoning in specification generation, bridging a critical gap left by prior benchmarks and reflecting the complexities of real-world software systems.

**Contribution 2.** We comprehensively evaluate 11 state-of-the-art LLMs on the constructed DAFNY-COMP (§4) benchmark across major providers; Figure 1 provides an overview. The results reveal a dramatic collapse in verification performance despite high syntactic accuracy: while models often produce syntactically well-formed outputs, verified success remains extremely low. With up to 8 attempts per problem, the best model reaches only 2% verification success, with most models below 1%, indicating that increased sampling provides only limited gains. Even after stronger test-time scaling via verifier-in-the-loop multi-turn self-refinement (up to 3 turns), verified rates improve but still remain far from saturation. Besides, on the non-chain DAG extension, we observe the same persistent Syntax–Verified gap with overall lower verified rates than in the chain setting, reinforcing that the core difficulty lies in coordinating interdependent contracts rather than in surface formatting.

**Contribution 3.** We conduct a systematic failure analysis on DAFNYCOMP to explain why verification collapses despite high syntactic well-formedness, and to move beyond aggregate success rates toward actionable diagnostics. Concretely, we manually inspect 900 representative failure cases from the chain split and distill three recurring failure modes that account for most breakdowns (§5): *specification fragility*, *implementation–proof misalignment*, and *reasoning instability*. We further provide illustrative examples and discussion for each category, showing how local-looking anno-

tations can be compositionally insufficient or internally inconsistent, thereby blocking end-to-end proofs. These findings highlight systemic obstacles to compositional formal verification beyond surface-level formatting errors, and position DAFNYCOMP as a diagnostic benchmark for tracking progress toward verifiable multi-component programs.

## 2 RELATED WORK

**Formal Verification Benchmarks.** Existing benchmarks for verifiable code generation can be categorized into two types. Single-function benchmarks, such as DAFNYBENCH (Loughridge et al., 2024) and MBPP-DFY (Misu et al., 2024), evaluate annotation completion within isolated methods, achieving moderate success rates (50-60%) but failing to capture inter-function dependencies. Interactive theorem proving benchmarks (miniCodeProps (Lohn & Welleck, 2024), FVAPPS (Dougherty & Mehta, 2025)) target proof synthesis in systems like Lean (De Moura et al., 2015) but require extensive manual validation and remain disconnected from practical programming. DAFNYCOMP bridges this gap by evaluating compositional specification generation—a prerequisite for scaling verification beyond toy programs to production systems. Unlike prior work, we explicitly construct multi-function programs with data dependencies, exposing the compositional reasoning deficit in current models (see Appendix C for detailed comparison).

**Dynamic Benchmark Generation.** Static benchmarks suffer from contamination and overfitting (Hu et al., 2025; Zhang et al., 2024). Dynamic generation techniques—creating new tasks, transforming problems, or perturbing reasoning structures (Zhu et al., 2024; 2023)—show promise in mathematics and logic but neglect formal verification's unique demands: specifications must be syntactically valid, semantically precise, and correct across all execution paths. Our synthesis pipeline addresses this by generating verifiable multi-function Dafny programs through controlled composition, ensuring both novelty and correctness while maintaining the semantic complexity that exposes compositional reasoning gaps.

**Compositional Reasoning in LLMs.** The ability to systematically combine simpler units into correct larger structures remains a frontier challenge (Li et al., 2024; Dziri et al., 2023). While progress exists in natural language and symbolic domains, formal verification imposes stricter demands: specifications must preserve invariants across components and ensure global correctness. Existing training paradigms favor pattern matching over principled proof construction. By requiring models to generate specifications bridge function boundaries with explicit data dependencies, DAFNYCOMP provides the first diagnostic benchmark for compositional reasoning in formal verification.

## 3 BENCHMARK CONSTRUCTION

DAFNYCOMP synthesizes 400 verified multi-function programs through a two-stage pipeline (Figure 2), including program assembly (§3.1) and formal translation (§3.2), which bridges the gap between practical Python implementations and verification-oriented Dafny specifications. Within this pipeline, program assembly ensures the construction of compositional Python programs with functional correctness, while specification translation with refinement ensures the quality and reliability of the resulting data. We also provide format details of evaluation tasks (§3.3) and the key characteristics of the benchmark (§3.4). **Unless stated otherwise, our primary Pass@k evaluation focuses on the chain split (300 programs), and we additionally include a non-chain DAG extension set (100 programs) to broaden topology diversity.**

### 3.1 PROGRAM ASSEMBLY

We construct compositional programs by systematically combining functions from LEETCODE-DATASET (Xia et al., 2025), selected for their algorithmic depth and verification challenges.

**Function Selection.** We filter the corpus using McCabe's cyclomatic complexity (McCabe, 1976) as a proxy for verification difficulty, retaining only functions with complexity >5 (around top 30% of the dataset) and at least 10 lines of code. This threshold ensures non-trivial control flow—loops with complex termination conditions, nested conditionals, recursive patterns—that stress specification generation. For tractability, we restrict to single-input/single-output functions, yielding 1,847 candidate functions.

**Compositional Strategy.** Following Hu et al. (2025), we primarily employ chain-based composition where each function's output feeds the next function's input, creating explicit data dependencies

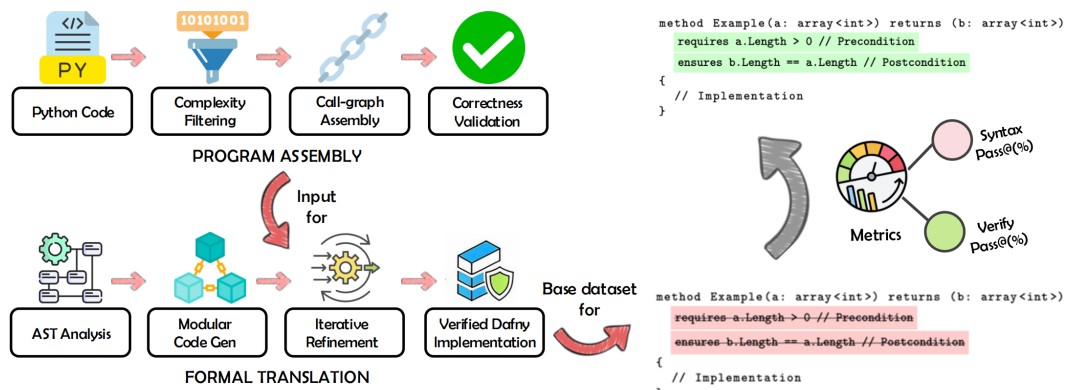

Figure 2: Two-stage benchmark synthesis: (1) Assembly combines independent Python functions with controlled data flow, ensuring algorithmic complexity while maintaining tractability; (2) Formal translation converts to verified Dafny through incremental AST-guided transformation.

while keeping the assembly procedure controlled. To broaden topology diversity beyond chains, we additionally instantiate 10 non-chain DAG topology templates and synthesize a 100-program DAG extension set, covering common branching patterns in acyclic call graphs (Appendix D lists the templates, illustrating the extensibility of our assembly to broader acyclic call graphs). After composition, we further identify the minimal set of shared import dependencies across the combined Python functions. This step is essential because the original LEETCODEDATASET often relies on broad `import *` statements, which obscure library ownership and names. Without explicit mappings, Dafny cannot interpret external libraries, preventing the synthesis of intermediate functions to replace missing third-party features. We generate programs with 2–5 functions, exploring multiple permutations since function ordering affects both data flow and verification complexity.

**Validation Pipeline.** After composition, the resulting Python code is subjected to a three-stage validation pipeline, which filters candidates before their use in Section 3.2.

- (**i**) *Type checking via constraint propagation*: We statically infer candidate types and shapes for each function's inputs and outputs and propagate these constraints along the composition chain. This pass rejects compositions with incompatible interfaces (e.g., scalar-sequence or element-type mismatches) and flags violations of simple value constraints inferred from guards (such as non-negativity or length bounds). The result is a set of compositions whose interfaces are consistent end-to-end, providing a reliable basis for subsequent translation and verification.

- (**ii**) *Formatting standardization*: We apply a deterministic rewriter, implemented with tools such as `Black` and `isort`, to normalize code style, including indentation, whitespace, line breaking, and import organization. Canonicalizing these incidental variations yields stable, diff-friendly artifacts and reduces prompt variance in later stages. This step preserves semantics while producing uniform program layouts that are easier to parse, translate, and verify.

- (**iii**) *Test validation*: We execute each composed program against the reference unit tests from LEETCODEDATASET to confirm functional correctness and basic executability. We construct unit tests for composed functions by intersecting the input–output constraints of their constituent functions. If the intersection is empty, the sample is discarded. Programs that raise runtime exceptions, fail assertions, or produce incorrect outputs are discarded, ensuring only behaviorally sound compositions advance. This filtering isolates verification challenges to specification and reasoning rather than implementation errors during the Dafny translation stage.

Following this procedure, we obtain 1,200 valid Python programs with 2–5 functions.

## 3.2 FORMAL TRANSLATION

We translate validated Python compositions into Dafny implementations with formal guarantees, focusing on the verification-oriented aspects of the benchmark here.

**Translation Challenges.** Direct end-to-end translation from Python to Dafny proved largely ineffective, with empirical success rates below 5%. The core difficulty lies in Dafny's demand for

explicit specifications, invariants, and termination arguments—semantic elements absent in Python. This semantic gap makes single-pass translation infeasible for non-trivial programs.

**Incremental Pipeline.** Inspired by Wen et al. (2024), we adopt an incremental approach: the abstract syntax tree (AST) of each Python program is decomposed into function- or control-structure–level fragments. Each fragment is translated into Dafny and immediately verified, localizing errors to the smallest possible unit. Verified fragments are then progressively reassembled according to the AST hierarchy, culminating in a complete Dafny program. Importantly, although translation proceeds incrementally, the Python program must be composed in its entirety before it can be executed. Whole-program composition in Python provides two benefits: (**i**) Python's explicit AST nodes and mature tooling make program assembly more reliable and transparent; and (**ii**) having a coherent Python blueprint ensures that the incremental Dafny translation preserves global logical relationships, rather than producing isolated fragments that fail to compose. Thus, whole-program composition and incremental translation are complementary design choices. To further improve reliability, each candidate Dafny program undergoes up to 10 refinement iterations, where specifications are strengthened in response to verifier feedback (e.g., adding loop invariants, refining postconditions...). We select CLAUDE-4-SONNET for the entire synthesis and refinement process; see Appendix E for success rate comparisons and Appendix F for the exact prompts. **To mitigate potential information leakage arising from evaluating models from the same family used in synthesis, we do not include Claude-family models in the downstream comparative evaluation.**

In total, the pipeline ultimately yields 564 verified Dafny programs from 1,200 attempts (corresponding to an overall 47% success rate). Translation synthesis errors primarily arise from incomplete specifications (31%), type inference errors (22%), timeouts (18%), and irreconcilable semantic gaps (29%). From these, we retain 300 chain-structured programs carefully balanced across complexity levels (100 each with 2–3, 3–4, and 4–5 functions) as our primary chain split. In addition, we apply the same assembly and formal-translation pipeline to non-chain DAG compositions and obtain 100 verified DAG-structured programs as an extension set, yielding 400 benchmarks in total. Given that DAFNYCOMP is automatically synthesized and intended to stress compositional verification, it is crucial that the resulting instances are both novel and not inadvertently recoverable from existing benchmarks or corpora. To ensure evaluation integrity, we conduct a thorough contamination analysis against the source Python dataset used by MBPP-DFY (Misu et al., 2024). The results in Appendix G provide strong evidence that our test set is indeed free from bias due to data overlap.

## 3.3 EVALUATION TASK FORMAT

We adopt a specification reconstruction task inspired by Loughridge et al. (2024). Still, with a crucial difference: rather than removing all annotation statements, we only strip away the contract clauses (`requires`, `ensures`, `reads`, `modifies`, `decreases`) that appear before the opening brace of each `method` or `function`. LLMs to be evaluated are then required to regenerate these specifications to enable verification. This design isolates the challenge of reconstructing cross-function contracts from implementation concerns, focusing evaluation on whether models can generate specifications that capture emergent correctness properties across component boundaries. Unlike annotation completion tasks that permit purely local reasoning, our multi-function programs require understanding how data flows and invariants propagate through compositions. We use a unified prompt and standardized compute settings across all evaluations (Appendix H and Appendix I).

## 3.4 BENCHMARK STATISTICAL SUMMARY

The resulting benchmark comprises 400 mechanically verified Dafny test cases, including a 300-program chain split and a 100-program non-chain DAG extension set. Unless stated otherwise, the statistics reported in this section summarize the **chain split**, and the DAG extension is described separately. The benchmark jointly captures three key dimensions: compositional complexity from function-to-function call dependencies, algorithmic diversity across multiple categories, and verification challenges arising from the increased specification burden.

**Compositional Complexity.** Each program contains 2–5 functions (mean = 3.2) with an average of 8.4 cross-function data dependencies, requiring models to reason about specification alignment across component boundaries. Unlike single-function benchmarks where specification generation is largely local, our programs demand that preconditions of called functions be implied by postconditions of their callers—a requirement that introduces cascading verification challenges when specifications fail to propagate correctly.

**Algorithmic Diversity.** The benchmark spans 15 algorithmic categories with balanced representation: dynamic programming (18%), string manipulation (20%), number theory (15%), and graph algorithms (12%) constitute the primary categories, with the remainder distributed across sorting, searching, and combinatorial problems. Beyond the balance of individual categories, diverse permutations and combinations of these types yield composed programs with more intricate, layered structures. Consequently, the target of composing function calls is reinforced by the characteristics of the source dataset (LEETCODEDATASET (Xia et al., 2025)), which in turn induces composition at the level of algorithmic logic. This design ensures models must develop general compositional reasoning rather than memorizing category-specific patterns.

**Verification Challenges.** Every program is mechanically verified by Dafny 4.10.0, thereby providing ground-truth correctness. The median program requires 7 loop invariants and 4 assertions for verification, with 23% demanding intricate termination arguments via `decreases` clauses—a 3.5× increase in annotation density compared to DAFNYBENCH's average of 2 per program. This added specification burden reflects the extra complexity of compositional verification, creating a graduated challenge that pinpoints where current models shift from local reasoning to compositional failure.

## 4 EXPERIMENTAL SETUP AND RESULTS

In this section, we enumerate the evaluation metrics (§4.1), LLM model selection for the benchmark (§4.2), and the corresponding evaluation results (§4.3).

### 4.1 METRICS

We evaluate two complementary aspects of model performance:

- **Syntax Correctness**: measures whether generated specifications parse successfully in Dafny. This baseline metric captures models' grasp of the formal language syntax.

- **Verified Rate**: measures the fraction of syntactically correct programs that pass Dafny's verifier—the ultimate test of semantic understanding. This metric is computed only over syntactically valid outputs, as verification requires parseable code.

**Independent sampling (Pass@$k$).** Following Chen et al. (2021), we report Pass@$k$ for $k \in \{1, 2, 4, 8\}$, measuring the overall probability of successfully solving a problem within $k$ attempts. Pass@1 provides a strict zero-shot baseline of immediate reasoning ability, whereas larger $k$ values further exploit additional test-time compute to improve success on compositional tasks (Snell et al., 2024). In this setting, Pass@8 provides a stable view of performance under moderate test-time compute beyond a single attempt.

**Verifier-in-the-loop multi-turn refinement.** In addition to independent sampling (Pass@k), we evaluate a stronger test-time scaling baseline based on verifier-in-the-loop multi-turn self-refinement. Starting from an initial generation (turn 1), if the output fails parsing or verification, we feed the model its previous output together with the verifier error message and request a revision while keeping the method signatures and declared contracts unchanged. We repeat this process for up to $T = 3$ turns, running the Dafny verifier once per turn under the same timeout, and report Syntax Correct Rate@turn$t$ and Verified Rate@turn$t$ for $t \in \{1, 2, 3\}$. Due to the cost of repeated generation and verification, we report multi-turn results on a representative subset of models.

### 4.2 MODEL SELECTION

We evaluate 11 frontier models spanning four major model families, chosen for their demonstrated strength in code generation and general reasoning:

- **OpenAI**: GPT-4O (Hurst et al., 2024), GPT-4.1 (OpenAI, 2024), O4-MINI (OpenAI, 2025)

- **Google**: GEMINI-2.5-PRO, GEMINI-2.5-FLASH (Google DeepMind, 2025)

- **DeepSeek**: DEEPSEEK-R1 (Guo et al., 2025), DEEPSEEK-V3 (Liu et al., 2024), DEEPSEEK-V3.1 (DeepSeek-AI, 2025)

- **Alibaba**: QWEN2.5-CODER-32B (Hui et al., 2024), QWEN3-CODER-480B (Qwen Team, 2025a), QWQ-32B (Qwen Team, 2025b)

Table 1: **Main results on the chain split (300 problems) under independent sampling (Pass@k).** We report **Syntax Correct Rate** and **Verified Rate** (i.e., Pass@k) for $k \in \{1, 2, 4, 8\}$ under a unified prompt and fixed verification procedure. Despite high syntactic well-formedness, verification success remains extremely low (best $2\%$ at Pass@8).

| Model | Syntax Correct Rate (%) | | | | Verified Rate (%) | | | |
|---|---|---|---|---|---|---|---|---|
| | @1 | @2 | @4 | @8 | @1 | @2 | @4 | @8 |
| **OpenAI Models** | | | | | | | | |
| GPT-4O | 94.33 | 98.67 | 99.33 | 99.67 | 0.33 | 0.33 | 0.33 | 0.33 |
| O4-MINI | 80.00 | 92.67 | 98.00 | 99.00 | 0.00 | 0.00 | 0.67 | 0.67 |
| GPT-4.1 | 59.00 | 69.67 | 79.33 | 86.33 | 0.00 | 0.00 | 0.00 | 0.00 |
| **Google Models** | | | | | | | | |
| GEMINI-2.5-FLASH | 54.00 | 64.00 | 81.00 | 89.67 | 0.00 | 0.00 | 0.00 | 0.00 |
| GEMINI-2.5-PRO | 69.00 | 81.00 | 91.67 | 96.00 | 0.00 | 0.33 | 0.67 | 2.00 |
| **DeepSeek Models** | | | | | | | | |
| DEEPSEEK-R1 | 85.67 | 95.33 | 98.33 | 99.00 | 0.33 | 0.33 | 0.33 | 0.33 |
| DEEPSEEK-V3 | 77.33 | 88.67 | 95.33 | 97.33 | 0.00 | 0.00 | 0.33 | 0.33 |
| DEEPSEEK-V3.1 | 54.67 | 72.33 | 83.33 | 92.00 | 0.00 | 0.00 | 0.00 | 0.67 |
| **Alibaba Models** | | | | | | | | |
| QWEN3-CODER-480B-A35B-INSTRUCT | 85.33 | 94.00 | 98.00 | 99.00 | 0.00 | 0.33 | 0.33 | 1.00 |
| QWEN2.5-CODER-32B-INSTRUCT | 62.00 | 74.67 | 85.00 | 89.00 | 0.00 | 0.33 | 0.33 | 0.67 |
| QWQ-32B | 46.67 | 61.33 | 78.00 | 91.00 | 0.00 | 0.00 | 0.00 | 0.00 |

Table 2: **Verifier-in-the-loop multi-turn self-refinement (representative models).** We report representative models (best-performing among evaluated multi-turn models per family under Verified Rate@T3) on the chain split (300 problems) and the DAG extension (100 problems) for readability; full results are provided in Appendix J.

| Model | Syntax Correct Rate (%) | | | Verified Rate (%) | | |
|---|---|---|---|---|---|---|
| | T1 | T2 | T3 | T1 | T2 | T3 |
| **Chain split** | | | | | | |
| DEEPSEEK-R1 | 85.67 | 94.33 | 95.00 | 0.33 | 5.67 | 9.67 |
| QWEN3-CODER-480B-A35B-INSTRUCT | 78.33 | 93.33 | 95.67 | 0.33 | 0.67 | 3.00 |
| O4-MINI | 37.33 | 92.00 | 96.00 | 0.00 | 2.00 | 9.67 |
| **DAG extension** | | | | | | |
| DEEPSEEK-V3.1 | 59.00 | 94.00 | 98.00 | 1.00 | 3.00 | 7.00 |
| QWEN3-CODER-480B-A35B-INSTRUCT | 69.00 | 95.00 | 97.00 | 0.00 | 1.00 | 4.00 |
| GPT-4.1 | 26.00 | 77.00 | 96.00 | 1.00 | 2.00 | 4.00 |

## 4.3 RESULTS AND DISCUSSION

Table 1 reports our main results on the chain split under independent sampling (Pass@$k$). To evaluate stronger test-time scaling beyond independent sampling, Table 2 reports a verifier-in-the-loop multi-turn self-refinement baseline on both the chain split and the DAG extension for representative models, with full results in Appendix J.

**Observation 1.** *Universal verification collapse despite syntactic mastery.* The most striking result is a large and consistent gap between syntax correctness and verification success across all evaluated models. At Pass@8, models achieve $\mu = 94.36\%$ (SD $= 4.86\%$) Syntax Correct Rate but only $\mu = 0.55\%$ (SD $= 0.58\%$) Verified Rate, yielding a 93.82% Syntax–Verified gap. This mismatch is not confined to a particular model family or scale: it appears across dense and MoE models, across general-purpose and code-specialized systems, and it persists even when increasing test-time compute via independent sampling (from Pass@1 to Pass@8). The universality of this failure suggests a systematic limitation in current LLMs' ability to coordinate interdependent contracts, rather than an issue that can be remedied by formatting fixes or modest increases in sampling.

**Observation 2.** *Compositionality induces disproportionate degradation from single-function verification.* In prior single-function verification benchmarks, frontier LLMs can achieve high syntax correctness ($> 99\%$) and moderate verified success ($> 58\%$). In contrast, on DAFNYCOMP's multi-function compositional tasks, verified success collapses to low single digits even under multiple attempts (average $0.55\%$, best $2.00\%$ at $k=8$). This degradation cannot be explained by additive difficulty alone: composing functions introduces interdependent contracts, where missing or slightly weakened postconditions in one component can fail to establish a callee's precondition downstream, cascading into end-to-end proof failures.

**Observation 3.** *Limited gains from sampling; multi-turn refinement helps but remains far from saturation.* Under independent sampling, verification gains saturate quickly: the average improvement from $k=4$ to $k=8$ is only $+0.27\%$, and $7/11$ models show no gain at all, whereas syntax correctness continues to improve with additional attempts. Under a stronger verifier-in-the-loop multi-turn baseline, Verified Rate improves substantially on the chain split (e.g., representative models reach up to $9.67\%$ at turn 3; Table 2), yet success remains far from saturation. Moreover, on the DAG extension, we observe a similarly persistent gap: even with refinement, Syntax Correct Rate@T3 remains high (up to $99\%$) while Verified Rate@T3 stays in the low single digits ($1$–$7\%$), with verified rates on DAG topologies lower on average than in the chain setting (Appendix J).

**Observation 4.** *Reasoning-specialized models show no clear advantage.* Models explicitly optimized for reasoning do not consistently outperform general-purpose models on DAFNYCOMP. For example, QWQ-32B still achieves $0\%$ verification at Pass@8, and DEEPSEEK-R1 remains below $1\%$ verification at Pass@8 despite its reinforcement-learning-based post-training. Overall, verification success is tightly clustered in the low single digits across diverse training objectives and model families. This null result is informative: it suggests that current reasoning-trace-based and reward-optimized approaches, as instantiated by today's frontier models, are not yet sufficient to reliably solve compositional verification on DAFNYCOMP.

## 5 FAILURE CASE ANALYSIS AND DISCUSSION

Table 3: Distribution of verification failure modes across 900 analyzed cases (3 best-performing models under our primary chain Pass@8 setting × 300 chain benchmarks) from DAFNYCOMP.

| Failure Mode | Frequency | % of Total | Primary Mechanism |
|---|---|---|---|
| Specification Fragility | 353/900 | 39.2 | Contract propagation failure |
| Implementation–Proof Misalignment | 195/900 | 21.7 | Independent generation pathways |
| Reasoning Instability | 127/900 | 14.1 | Inductive chain breakdown |
| Other (syntax, timeout, misc.) | 225/900 | 25.0 | Various |

The significant gap between syntax correctness and ultimate verification success demands a clear mechanistic explanation. We therefore conduct a systematic manual analysis of 900 failures drawn from the chain split (300 programs × the three best-performing models under our primary Pass@8 evaluation). Table 3 summarizes the distribution of failure modes over this analyzed subset; our goal is to distill recurring breakdown patterns rather than to estimate a global failure distribution across all models. We discuss each mode below and provide additional concrete examples in Appendix K.

### 5.1 SPECIFICATION FRAGILITY: THE DOMINO EFFECT

*Specification fragility*, the inability to generate contracts that remain valid across function compositions, accounts for most failures. Consider a representative case from DAFNYCOMP: a `digitSum` function correctly implemented but missing the postcondition `ensures result >= 0`. In isolation, this omission appears minor. In composition, it cascades—when `digitSum`'s output feeds a caller expecting non-negative input, verification fails globally despite both functions being locally correct. Note that this pattern recurs throughout our dataset: models generate specifications sufficient for local correctness but insufficient for compositional soundness. A `requires n >= 0` precondition absent from one function invalidates the entire pipeline's verification, even when each component individually passes most test cases. The fragility stems from a fundamental mismatch: LLMs learn specifications as local patterns rather than global contracts. They lack the architectural machinery to reason about how data constraints propagate through function calls—a capability essential for modular verification. More broadly, systems requiring compositional correctness—from

distributed systems protocols to smart contract verification—may face similar failures until models can reason about specification flow across component boundaries.

The first key takeaway insight about *specification fragility* is summarized below:

> **Takeaways (i̲)**: *LLMs often handle local specs but fail under composition. Missing contract propagation is the main cause of verification breakdowns.*

## 5.2 IMPLEMENTATION–PROOF MISALIGNMENT: THE INDEPENDENCE ASSUMPTION

The second failure mode points to a generation-level mismatch: models often produce implementations and specifications as if they were independent, rather than as coupled constraints. In 21.7% of the analyzed cases, syntactically valid code contradicts its own specifications. One striking example: a model generated `assert 0 >= 1;` within otherwise reasonable code, illustrating a recurring pattern of internally contradictory assertions rather than an isolated slip. More subtle misalignments prove equally fatal: loop invariants like `forall k :: 0 <= k < i ==> cnt[k] >= 0` appear plausible but fail verification because the implementation's array access patterns violate the stated bounds. This pattern is consistent with prior mechanistic analyses suggesting that transformer circuits often implement relatively local algorithmic behaviors rather than globally consistent proof construction (Olsson et al., 2022; Elhage et al., 2021) Current training paradigms further exacerbate this issue: models learn from code–specification pairs without explicit feedback on mutual consistency, yielding impressive syntax but brittle semantic alignment in composed settings.

We summarize the second key takeaway about *implementation-proof misalignment* as:

> **Takeaways (ii̲)**: *Code and specs are generated independently, leading to plausible but inconsistent invariants. Future training for this task should enforce better alignment.*

## 5.3 REASONING INSTABILITY: INDUCTION AS ACHILLES' HEEL

The third failure pattern, which we refer to as *reasoning instability*, highlights a major barrier in sustaining the inductive arguments required by formal verification. Formal proofs often require establishing a base case, maintaining invariants through iterative updates, and composing these guarantees across function boundaries. In DAFNYCOMP, models frequently fail to maintain this inductive chain: loop invariants that should accumulate state (e.g., `invariant res == stringToIntHelper(s[..i])`) break as models lose track of how program state evolves, and recursive functions often miss well-founded decreases measures or inductive strengthening needed to generalize beyond base cases. These failures suggest that, while LLMs can match locally plausible invariant templates, they often struggle to maintain globally consistent inductive reasoning, especially in compositional settings where relationships must be preserved across calls.

We summarize the third insight about *reasoning instability* as:

> **Takeaways (iii̲)**: *LLMs approximate base cases but fail to sustain inductive reasoning, exposing a structural gap in formal verification.*

## 6 LIMITATIONS AND FUTURE WORK

While DAFNYCOMP exposes fundamental limitations in compositional reasoning, we want to gently mention several constraints of our evaluation, which indicate some interesting future work.

- **Compositional Patterns.** Our primary evaluation focuses on acyclic chain-structured compositions for tractability, and the benchmark additionally includes a 100-program non-chain DAG extension. More complex patterns such as cyclic graphs, mutual recursion, or shared-state-heavy dependencies remain out of scope. Extending to these settings would require addressing synthesis and verification tractability in the presence of cyclic dependencies.

- **Specification Types.** Our benchmark tests functional correctness (preconditions, postconditions, invariants) but not liveness properties, resource bounds, or security policies. These orthogonal

concerns—e.g., proving memory consumption remains constant across compositions—require different verification techniques and evaluation metrics.

- **Data Scarcity.** The core challenge may be training data availability. Repositories contain only a few verified multi-function programs with compositional specifications. Synthetic data generation or bootstrapped program synthesis could address this gap, at scale and with stronger coverage, although ensuring semantic diversity remains a challenge.

## 7 CONCLUSION

We introduce DAFNYCOMP, the first benchmark specifically designed to evaluate the generation of compositional specifications for formal verification. DAFNYCOMP contains 400 mechanically verified multi-function Dafny programs, including a 300-program chain split and a 100-program non-chain DAG extension. We evaluate 11 frontier LLMs under a unified prompting and fixed verification protocol. Our results reveal a stark Syntax–Verified gap: while models achieve high syntax correctness, verified success on compositional tasks remains extremely low (average 0.55%, best 2.00% at Pass@8 on the chain split). Stronger test-time scaling via verifier-in-the-loop multi-turn self-refinement improves success but remains far from saturation, and the difficulty persists on the DAG extension with lower verified rates. Through manual failure analysis, we identify three recurring breakdown patterns—specification fragility, implementation–proof misalignment, and reasoning instability—that systematically hinder end-to-end compositional verification. Overall, DAFNY-COMP provides a diagnostic target for tracking progress in verifiable code generation and highlights that bridging local correctness to compositional verification remains a key open challenge. We release the benchmark, evaluation framework, and synthesis pipeline to support future research.

## ACKNOWLEDGMENT

This work is supported by the HKUST startup grant R9895 from CSE; RGC-ECS project 26218024; RGC-NSFC project CRS_HKUST601/24. We thank the support from National Supercomputer Center in Guangzhou Nansha Sub-Center.

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

## A  THE USE OF LLMs IN WRITING

During the preparation of this manuscript, we employed a large language model (OpenAI GPT-5) to assist with language refinement and editorial improvements. Specifically, the LLM was used to enhance sentence fluency, improve clarity of expression, and ensure consistency with academic writing conventions. The tool was applied exclusively for linguistic polishing—all research design, experimental work, data analysis, and core intellectual contributions remain entirely original.

## B  INTRODUCTION TO DAFNY

Dafny (Leino, 2010), developed at Microsoft Research, is a verification-oriented programming language specifically designed to support formal reasoning about software. Unlike conventional languages where correctness is primarily assessed through testing, Dafny integrates an automated program verifier directly into the development workflow, enabling developers to construct code that is mathematically proven to satisfy its specifications. This approach shifts the discovery of defects from the testing phase to the design and implementation phases, thereby improving software reliability.

A distinctive feature of Dafny is that specifications are treated as first-class citizens. Methods can be annotated with preconditions, postconditions, and logical properties that describe intended behavior. For example:

```
method Example(a: array<int>) returns (b: array<int>)
  requires a.Length > 0              // Precondition
  ensures b.Length == a.Length       // Postcondition
  ensures forall i :: 0 <= i < b.Length ==> b[i] >= 0  // Property
{
  // Implementation
}
```

The Dafny verifier relies on automated theorem proving (via Z3 solver (De Moura & Bjørner, 2008)) to ensure that implementations conform to these specifications, providing mathematical certainty about program behavior. Crucially, the ability to reason about the composition of verified components determines whether verification can scale from toy examples to real-world systems. Without compositional reasoning, verification remains confined to small, isolated programs rather than production-level software.

## C  AUTOMATED THEOREM PROVING

A complementary line of work contrasts automated verification frameworks with interactive theorem proving (ITP) systems. Languages such as Dafny and Verus rely on SMT solvers to discharge proof obligations, requiring only lightweight annotations (e.g., invariants, assertions). This design lowers the barrier to entry but is constrained by the solver's limited reasoning scope and opaque failure modes. In contrast, ITPs such as Lean expose every proof step explicitly, enabling iterative refinement and error diagnosis. Recent studies even show that LLMs can generate competition-level mathematical proofs in Lean. However, existing Lean-based benchmarks (e.g., miniCodeProps, FVAPPS) either focus narrowly on proof synthesis or lack human validation. By comparison, Dafny offers a more balanced environment for benchmarking LLMs: it combines code, specifications, and automated verification in a way that remains close to mainstream programming practice.

**How Dafny Works and Its Core Strengths.** Dafny's approach stems from its verification-aware design. Developers embed formal specifications, such as preconditions, postconditions, and loop invariants, directly within the code (Leino, 2010). These specifications are not merely comments; they are integral components checked by the built-in verifier. The verifier translates Dafny code and its specifications into an intermediate verification language, Boogie, which then generates proof obligations. These obligations are processed by an SMT solver (e.g., Z3) to prove their validity. If all obligations are proven, the code is confirmed to be correct according to its specifications. If a proof fails, Dafny provides precise feedback on the inconsistencies. This methodology supports correctness by construction, helping to reduce common errors like null pointer dereferences or array out-of-bounds access (Poesia et al., 2024). Once verified, Dafny code can be translated into mainstream languages such as Python for execution (Li et al., 2025).

**Dafny vs. Python: A Fundamental Difference in Approach.** To understand Dafny's position, it's useful to compare it with a widely used language like Python. While both are effective, their fundamental design philosophies and primary objectives differ, as shown in Table 4.

| Feature | Dafny | Python |
|---|---|---|
| Year Introduced | 2010 (Microsoft Research) | 1991 (Guido van Rossum) |
| Type System | Static typing, compile-time checks | Dynamic typing, run-time checks |
| Formal Verification | Yes — built-in contracts and proofs | No — only basic `assert` |
| Main Use | Verified algorithms, critical systems | General-purpose programming |
| Execution Model | Compiled with verification | Interpreted (e.g., CPython) |

Table 4: Key differences between Dafny and Python.

In summary, Dafny offers a distinct approach to software development by integrating formal verification into the language itself. While Python excels in agile development and broad applicability, Dafny is particularly suited for domains where software correctness and formal guarantees are critical. For more, please refer to the Dafny official website[1].

## D  DAG TOPOLOGY TEMPLATES

To broaden topology diversity beyond chains, we instantiate 10 acyclic DAG templates and synthesize 100 DAG-structured programs (10 per template) as an extension set in DAFNYCOMP.

| ID | Topology description |
|---|---|
| 1 | 4 nodes, root splits into two, one branch splits again |
| 2 | 4 nodes, root splits into three |
| 3 | 4 nodes, chain with a branch at the second node |
| 4 | 5 nodes, chain then splits into three |
| 5 | 5 nodes, root splits into four |
| 6 | 5 nodes, root splits into three, leftmost branch splits again |
| 7 | 5 nodes, chain with a branch at the second node, both branches split again |
| 8 | 5 nodes, root splits into two, left branch splits again |
| 9 | 5 nodes, root splits into two, both branches are chains of two |
| 10 | 5 nodes, root splits into two, one branch is a chain of three |

Table 5: DAG topology templates used to instantiate the 100-program DAG extension set.

---

[1]`https://dafny.org/dafny/OnlineTutorial/guide`

# E  MODEL SELECTION FOR FORMAL TRANSLATION

Concretely, we randomly sampled 100 test cases from the 1,200 instances obtained after our assembly procedure to evaluate each model with identical prompts and decoding settings. We define success rate as the fraction of generations whose specifications are valid under the Dafny verifier.

| Model | Success rate (%) |
|---|---|
| Claude-4-Sonnet-20250514 | 58.00 |
| claude-3.5-sonnet-20241022 | 55.00 |
| gpt-3.5-turbo | 45.00 |
| gpt-4o | 31.00 |
| gpt-4o-mini | 41.00 |
| o1 | 36.00 |
| o1-mini | 33.00 |
| o3-mini | 37.00 |
| gemini-2.0-flash | 38.00 |

Table 6: Success rates for the formal translation step.

As shown, we ultimately chose Claude-4-Sonnet-20250514 because it yielded the highest rate of valid specifications on these cases. This aligns with our design goal to prioritize the quality of generated specifications used downstream in our experiments.

## F  PROMPTS FOR SYNTHESIS

The prompt templates used for annotating data with Claude-4 Sonnet are shown in the following boxes.

---

**Prompt for Initial Dafny Code Generation**

**SYSTEM**
You are an expert AI assistant that writes Dafny programs. You excel at writing code with formally verified correctness, providing precise preconditions and postconditions, and finding the appropriate loop invariants to ensure all verification conditions are met.

**TASK**
Below is the Python code:

```python
<python_code>
```

Please translate this Python code into Dafny, ensuring:
1. **Method Signatures**: Each piece of functionality should be expressed as a Dafny method (or set of methods) with a well-defined signature.
2. **Preconditions**: Clearly state any 'requires' clauses for each method (e.g., array length constraints, non-null references, numeric domain restrictions, etc.).
3. **Postconditions**: State the logical guarantees about the returned values or final state as 'ensures' clauses (e.g., correctness of returned results, absence of side effects, etc.).
4. **Verification Details**: Include all necessary loop invariants (or other verification hints) so Dafny can prove the postconditions, along with a brief explanation. For example: - Explain how you chose your invariants. - Describe how they ensure the correctness of the loop.

Return the final Dafny code as a self-contained snippet that can be verified by Dafny as-is, with a short explanation of how it connects to the original Python functionality.

**AI ASSISTANT**
<The LLM's generated Dafny code with specifications here.>

---

**Dynamic Debugging Prompt for Code Generation**

**SYSTEM**
You are an expert AI assistant that writes and debugs Dafny programs. You excel at diagnosing and fixing verification errors based on Dafny solver messages, while maintaining correct preconditions, postconditions, and loop invariants.

**TASK**
Below is the Python code:

```python
<python_code>
```

And the Dafny code you previously provided (which I tried to verify):

```dafny
<main_spec>
```

---

I ran `dafny verify *.dfy` and received this error message:

```
<dafny_analysis_result>
```

Can you please fix the main function specification so that it parses successfully? Output the corrected main function specification only, without any other text.

**AI ASSISTANT**
<The LLM's generated Dafny code with specifications here.>

## G   DATA CONTAMINATION ANALYSIS

To validate the novelty of DAFNYCOMP, we conducted a rigorous data contamination analysis against the widely-used MBPP dataset (Austin et al., 2021), used to assess contamination in Python source data. We confirm that our benchmark source data shows no significant overlap, ensuring model performance reflects genuine reasoning capabilities rather than memorization.

Our analysis, focusing solely on code, employs two standard metrics: **Exact Match** to detect verbatim copies, and **n-gram Jaccard Similarity** to identify structurally similar code. We performed this analysis under four distinct configurations, the results of which are summarized in Table 7.

Across all scenarios, we found **zero exact matches**. The n-gram Jaccard similarity remains negligible, peaking at a mere 0.0078 even under the most aggressive settings. These findings provide strong evidence that DAFNYCOMP is free from training data contamination.

Table 7: Summary of Data Contamination Analysis. The table shows results for four testing configurations: **A (Conservative)** with minimal preprocessing; **B (Default)** with moderate preprocessing; **C (Aggressive)** with extensive preprocessing; and **D (Holistic)** for a structure-level check. Across all configurations, results show zero exact matches and negligible n-gram similarity when comparing DAFNYCOMP source data against MBPP, confirming the benchmark's integrity.

| Analysis Configuration | N-gram ($n$) | Exact Overlap | Max Jaccard | vs. sanitized-mbpp |
|---|---|---|---|---|
| A: Conservative | 15 | 0 | 0.000078 | 0 |
| B: Default | 11, 13, 15 | 0 | 0.000389 | 0 |
| C: Aggressive | 9, 11, 13 | 0 | 0.007757 | 0 |
| D: Holistic | 11, 13, 15 | 0 | 0.000234 | 0 |

## H   PROMPT FOR EVALUATION

The prompt template used for evaluation is shown in the following box. Note that all model outputs are used directly for Dafny verification.

**Evaluation Prompt for Dafny Specification Generation**

**SYSTEM**
You are an expert in Dafny. You will be given tasks dealing with Dafny programs including precise annotations. You should only return code body in all circumstances. No text is allowed.

**TASK**
Given a Dafny program with function signature, preconditions, postconditions, and code, but with annotations missing. Please return a complete Dafny program with the strongest possible annotation (loop invariants, assert statements, etc.) filled back in. Do not explain or output any text. If you have to explain, put all explanations in comments form. There

should only be code body in your output. Please use exactly the same function signature, preconditions, and postconditions. Do not ever modify the given lines.
Below is the program:

```dafny
<dafny_program_with_missing_annotations>
```

**AI ASSISTANT**

```dafny
<The LLM's generated Dafny code with specifications
here.>
```

## I    COMPUTE SETTINGS

Details of Compute Settings

**Token budgets:**
- No hard constraint is imposed on the number of generated tokens.
- For each model, `max_output_tokens` (or equivalent) is set to the **largest value allowed by the provider**, so that long reasoning traces are not truncated.
- Many evaluated systems are reasoning-oriented (e.g., R1) and may follow long, iterative reasoning trajectories, so token usage is largely emergent and model-dependent.

**Inference sampling:**
- Default decoding for most models: temperature $= 0.7$, top-p $= 0.8$.
- O4-MINI: provider default configuration, with fixed temperature $= 1$ and no top-p parameter.

**Verifier retry policy:**
- Exactly one verification attempt per output.
- SMT solver timeout: $60$ s.
- If the solver does not finish within $60$ s, the instance is treated as unsolved (not counted as a success).

## J    VERIFIER-IN-THE-LOOP MULTI-TURN SELF-REFINEMENT (FULL RESULTS)

Table 8: **Full multi-turn results on the chain split (300 problems).** We report **Syntax Correct Rate** and **Verified Rate** (end-to-end verifier success) at each refinement turn $T \in \{1, 2, 3\}$. All values are percentages.

| | Syntax Correct Rate (%) | | | Verified Rate (%) | | |
|---|---|---|---|---|---|---|
| Model | T1 | T2 | T3 | T1 | T2 | T3 |
| DEEPSEEK-R1 | 85.67 | 94.33 | 95.00 | 0.33 | 5.67 | 9.67 |
| DEEPSEEK-V3.1 | 49.00 | 91.67 | 96.00 | 0.00 | 1.67 | 5.33 |
| DEEPSEEK-V3 | 74.33 | 95.00 | 96.00 | 0.00 | 2.67 | 5.00 |
| QWQ-32B | 44.67 | 56.67 | 61.67 | 0.00 | 0.00 | 1.00 |
| QWEN2.5-CODER-32B-INSTRUCT | 59.67 | 62.33 | 63.67 | 0.33 | 0.33 | 0.67 |
| QWEN3-CODER-480B-A35B-INSTRUCT | 78.33 | 93.33 | 95.67 | 0.33 | 0.67 | 3.00 |
| GPT-4.1 | 46.67 | 86.67 | 92.67 | 0.00 | 1.33 | 3.33 |
| GPT-4O | 94.00 | 98.33 | 98.67 | 0.33 | 2.00 | 4.67 |
| O4-MINI | 37.33 | 92.00 | 96.00 | 0.00 | 2.00 | 9.67 |

Table 9: **Full multi-turn results on the DAG extension (100 problems).** We report **Syntax Correct Rate** and **Verified Rate** (end-to-end verifier success) at each refinement turn $T \in \{1, 2, 3\}$. All values are percentages.

| | Syntax Correct Rate (%) | | | Verified Rate (%) | | |
|---|---|---|---|---|---|---|
| Model | T1 | T2 | T3 | T1 | T2 | T3 |
| DEEPSEEK-R1 | 52.00 | 79.00 | 79.00 | 1.00 | 4.00 | 6.00 |
| DEEPSEEK-V3.1-TERMINUS | 59.00 | 94.00 | 98.00 | 1.00 | 3.00 | 7.00 |
| DEEPSEEK-V3 | 67.00 | 96.00 | 98.00 | 0.00 | 4.00 | 4.00 |
| QWQ-32B_DAG | 29.00 | 42.00 | 49.00 | 0.00 | 0.00 | 2.00 |
| QWEN2.5-CODER-32B-INSTRUCT | 51.00 | 62.00 | 63.00 | 0.00 | 1.00 | 1.00 |
| QWEN3-CODER-480B-A35B-INSTRUCT | 69.00 | 95.00 | 97.00 | 0.00 | 1.00 | 4.00 |
| GPT-4.1 | 26.00 | 77.00 | 96.00 | 1.00 | 2.00 | 4.00 |
| GPT-4O | 69.00 | 99.00 | 99.00 | 0.00 | 0.00 | 2.00 |
| O4-MINI | 22.00 | 76.00 | 93.00 | 0.00 | 1.00 | 4.00 |

## K   REAL EXAMPLES FROM BENCHMARK DATA

### K.1   CHALLENGE 1: FORMAL SPECIFICATION SYNTHESIS

The following examples demonstrate failures in generating correct formal specifications, particularly preconditions and postconditions.

### K.1.1   EXAMPLE 1.1: POSTCONDITION FAILURE (ORG_INPUT_ID: 2, SELF_ID: 20001)

**LLM Response:**

```
method digitSum(n: int) returns (sum: int)
    requires n >= 0
    ensures sum >= 0
{
    var temp := n;
    sum := 0;
    while temp > 0
        invariant temp >= 0
        invariant sum >= 0
    {
        sum := sum + (temp % 10);
        temp := temp / 10;
    }
}

method countLargestGroup_1399(n: int) returns (result: int)
    requires n >= 0
    ensures result >= 1
    ensures result <= 1000
{
    var counts := new int[50];
    var i := 0;
    while i < 50
        invariant 0 <= i <= 50
        invariant forall j :: 0 <= j < i ==> counts[j] == 0
    {
        counts[i] := 0;
        i := i + 1;
    }
    var maxCount := 0;
    var groupsWithMaxCount := 0;
    i := 1;
    while i <= n
        invariant 1 <= i <= n + 1
        invariant maxCount >= 0
        invariant groupsWithMaxCount >= 0
        invariant maxCount > 0 ==> groupsWithMaxCount >= 1
        invariant forall j :: 0 <= j < 50 ==> counts[j] >= 0
    {
        var ds := digitSum(i);
        if ds < 50 {
            counts[ds] := counts[ds] + 1;
            if maxCount < counts[ds] {
                maxCount := counts[ds];
                groupsWithMaxCount := 1;
            } else if maxCount == counts[ds] {
                groupsWithMaxCount := groupsWithMaxCount + 1;
            }
        }
        i := i + 1;
    }
    result := groupsWithMaxCount;
    if result == 0 {
        result := 1;
    }
    assert result >= 1;
```

```
        if result > 1000 {
            result := 1000;
        }
}

method main_3node_2(o: int) returns (result: int)
    requires o >= 0
    ensures result == -1 || result >= 2
{
    var o1 := countLargestGroup_1399(o);
    var o2 := sumOfMultiples_2652(o1);
    if o2 >= 1 {
        var o3 := nextGreaterElement_556(o2);
        result := o3;
        if result != -1 && result <= 1 {
            result := 2;
        }
    } else {
        result := -1;
    }
}
```

**Dafny Error:**

```
ex.dfy(137,0): Error: a postcondition could not be proved on this return
    path
ex.dfy(136,25): Related location: this is the postcondition that could
    not be proved
```

### K.1.2 EXAMPLE 1.2: MISSING PRECONDITION (ORG_INPUT_ID: 6, SELF_ID: 60005)

**LLM Response:**

```
method DigitSum(n: int) returns (sum: int)
    requires n >= 0
    ensures sum >= 0
{
    var num := n;
    sum := 0;
    while num > 0
        invariant sum >= 0
        invariant num >= 0
        decreases num
    {
        sum := sum + (num % 10);
        num := num / 10;
    }
}

method countLargestGroup_1399(n: int) returns (result: int)
    requires n >= 1
    ensures result >= 1
{
    var digitSumCounts := map[];
    var maxCount := 0;
    var i := 1;
    while i <= n
        invariant 1 <= i <= n + 1
        invariant maxCount >= 0
        invariant forall k :: k in digitSumCounts ==> digitSumCounts[k]
            >= 0
        decreases n - i + 1
    {
        var digitSum := DigitSum(i);
        var currentCount := if digitSum in digitSumCounts then
            digitSumCounts[digitSum] else 0;
        digitSumCounts := digitSumCounts[digitSum := currentCount + 1];
        if currentCount + 1 > maxCount {
            maxCount := currentCount + 1;
        }
        i := i + 1;
    }
    result := 0;
    var possibleSum := 0;
    while possibleSum <= 9 * 4
        invariant possibleSum >= 0
        invariant result >= 0
        invariant forall k :: k in digitSumCounts ==> digitSumCounts[k]
            >= 0
        decreases 9 * 4 - possibleSum + 1
    {
        if possibleSum in digitSumCounts && digitSumCounts[possibleSum]
            == maxCount {
            result := result + 1;
        }
        possibleSum := possibleSum + 1;
    }
    if result == 0 {
        result := 1;
    }
}

method main_4node_4(o: int) returns (result: int)
    ensures result >= 1
{
```

```
    var o1 := nextGreaterElement_556(o);
    var o2: int;
    if o1 == -1 {
        o2 := 1;
    } else if o1 > 10000 {
        o2 := 10000;
    } else {
        o2 := o1;
    }
    var o3 := countLargestGroup_1399(o2);
    var clampedO3: int;
    if o3 > 1000000000000000 {
        clampedO3 := 1000000000000000;
    } else if o3 < -1000000000000000 {
        clampedO3 := -1000000000000000;
    } else {
        clampedO3 := o3;
    }
    var o4 := smallestNumber_2165(clampedO3);
    var finalInput: int;
    if o4 < 1 {
        finalInput := 1;
    } else if o4 > 15 {
        finalInput := 15;
    } else {
        finalInput := o4;
    }
    result := countArrangement_526(finalInput);
}
```

**Dafny Error:**

```
ex.dfy(177,36): Error: a precondition for this call could not be proved
```

### K.2 CHALLENGE 2: ENSURING CODE CORRECTNESS

These examples show implementation issues where the code logic doesn't satisfy the specifications.

#### K.2.1 EXAMPLE 2.1: ARRAY BOUNDS ERROR (ORG_INPUT_ID: 1, SELF_ID: 10002)

**LLM Response:**

```
method smallestNumber_2165(num: int) returns (result: int)
{
    var neg := num < 0;
    var n := if num < 0 then -num else num;
    var cnt := new int[10];
    var i := 0;
    while i < 10
        invariant 0 <= i <= 10
        invariant cnt.Length == 10
        invariant forall k :: 0 <= k < i ==> cnt[k] >= 0
    {
        cnt[i] := 0;
        i := i + 1;
    }
    // ... rest of implementation
}

method main_5node_8(o: int) returns (result: int)
{
    var o1 := smallestNumber_2165(o);
    var o2 := clumsy_1006(if o1 <= 10000 then o1 else 10000);
    var o3 := lastRemaining_390(o2);
    var o4 := getMoneyAmount_375(if o3 <= 200 then o3 else 200);
    var o5 := numberOfWays_3183(if o4 <= 100000 then o4 else 100000);
    result := o5;
}
```

**Dafny Error:**

```
ex.dfy(44,30): Error: this loop invariant could not be proved on entry
Related message: loop invariant violation
```

### K.2.2 EXAMPLE 2.2: ASSERTION FAILURE (ORG_INPUT_ID: 6, SELF_ID: 60001)

**LLM Response:**

```
method DigitSum(n: int) returns (sum: int)
    requires n >= 0
    ensures sum >= 0
{
    var num := n;
    sum := 0;
    while num > 0
        invariant sum >= 0
        invariant num >= 0
        decreases num
    {
        sum := sum + (num % 10);
        num := num / 10;
    }
}

method countLargestGroup_1399(n: int) returns (result: int)
    requires n >= 1
    ensures result >= 1
{
    var digitSumCounts := map[];
    var maxCount := 0;
    var i := 1;
    while i <= n
        invariant 1 <= i <= n + 1
        invariant maxCount >= 0
        invariant forall k :: k in digitSumCounts ==> digitSumCounts[k] >
            0
        decreases n - i + 1
    {
        var digitSum := DigitSum(i);
        var currentCount := if digitSum in digitSumCounts then
            digitSumCounts[digitSum] else 0;
        digitSumCounts := digitSumCounts[digitSum := currentCount + 1];
        if currentCount + 1 > maxCount {
            maxCount := currentCount + 1;
        }
        i := i + 1;
    }

    result := 0;
    var possibleSum := 0;
    while possibleSum <= 9 * 4
        invariant possibleSum >= 0
        invariant result >= 0
        decreases 9 * 4 - possibleSum + 1
    {
        if possibleSum in digitSumCounts && digitSumCounts[possibleSum]
            == maxCount {
            result := result + 1;
        }
        possibleSum := possibleSum + 1;
    }

    if result == 0 {
        result := 1;
    }
}

method main_4node_4(o: int) returns (result: int)
    ensures result >= 1
{
```

```
    var o1 := nextGreaterElement_556(o);
    var o2: int;
    if o1 == -1 {
        o2 := 1;
    } else if o1 > 10000 {
        o2 := 10000;
    } else {
        o2 := o1;
    }
    assert o2 >= 1;

    var o3 := countLargestGroup_1399(o2);
    var clampedO3: int;
    if o3 > 1000000000000000 {
        clampedO3 := 1000000000000000;
    } else if o3 < -1000000000000000 {
        clampedO3 := -1000000000000000;
    } else {
        clampedO3 := o3;
    }

    var o4 := smallestNumber_2165(clampedO3);
    var finalInput: int;
    if o4 < 1 {
        finalInput := 1;
    } else if o4 > 15 {
        finalInput := 15;
    } else {
        finalInput := o4;
    }
    assert finalInput >= 1;

    result := countArrangement_526(finalInput);
}
```

**Dafny Error:**

```
ex.dfy(181,4): Error: assertion might not hold
assert o2 >= 1;
```

### K.3    CHALLENGE 3: PROOF GENERATION AND FORMAL REASONING

Examples demonstrating failures in generating proper loop invariants and formal proofs.

#### K.3.1    EXAMPLE 3.1: INVARIANT NOT MAINTAINED (ORG_INPUT_ID: 0, SELF_ID: 6)

**LLM Response:**

```
method stringToInt(s: seq<char>) returns (result: int)
  requires |s| > 0
  requires forall i :: 0 <= i < |s| ==> '0' <= s[i] <= '9'
  ensures result >= 0
{
  var res := 0;
  var i := 0;
  while i < |s|
    invariant 0 <= i <= |s|
    invariant res >= 0
    invariant res == stringToIntHelper(s[..i])
  {
    var digit := digitToInt(s[i]);
    assert 0 <= digit <= 9;
    res := res * 10 + digit;
    i := i + 1;
  }
  return res;
}

method monotoneIncreasingDigits_738(n: int) returns (result: int)
  requires n >= 0
  ensures result >= 1
  ensures result <= 1000000000
{
  if n == 0 {
    return 1;
  }
  var s := intToString(n);
  var i := 1;
  while i < |s| && digitToInt(s[i-1]) <= digitToInt(s[i])
    invariant 1 <= i <= |s|
    invariant forall j :: 0 <= j < |s| ==> '0' <= s[j] <= '9'
  {
    i := i + 1;
  }
  if i < |s| {
    while i > 0 && i < |s| && digitToInt(s[i-1]) > digitToInt(s[i])
      invariant 0 <= i < |s|
      invariant forall j :: 0 <= j < |s| ==> '0' <= s[j] <= '9'
      decreases i
    {
      var prevDigit := digitToInt(s[i-1]);
      if prevDigit > 0 {
        s := s[i-1 := intToChar(prevDigit - 1)];
      }
      i := i - 1;
    }
    i := i + 1;
    while i < |s|
      invariant i <= |s|
      invariant forall j :: 0 <= j < |s| ==> '0' <= s[j] <= '9'
    {
      s := s[i := '9'];
      i := i + 1;
    }
  }
  var res := stringToInt(s);
  if res >= 1 && res <= 1000000000 {
```

```
      return res;
    } else {
      return 1;
    }
}

method main_3node_2(o: int) returns (result: int)
   requires o >= 0
   ensures result >= 0
{
  var o1 := smallestFactorization_625(o);
  if o1 == 0 {
    return 0;
  }
  var o2 := monotoneIncreasingDigits_738(o1);
  var o3 := minOperations_2571(o2);
  return o3;
}
```

**Dafny Error:**

```
ex.dfy(25,18): Error: this invariant could not be proved to be maintained
     by the loop
Related message: loop invariant violation
```

### K.3.2 EXAMPLE 3.2: RECURSION TERMINATION (ORG_INPUT_ID: 4, SELF_ID: 40003)

**LLM Response:**

```
function gcd_func(a: int, b: int): int
{
    if b == 0 then a else gcd_func(b, a % b)
}

method gcd(a: int, b: int) returns (result: int)
requires a > 0 && b >= 0
ensures result == gcd_func(a, b)
{
    var x := a;
    var y := b;
    while y != 0
        invariant x > 0 && y >= 0
        invariant gcd_func(a, b) == gcd_func(x, y)
        decreases y
    {
        var temp := x % y;
        x := y;
        y := temp;
    }
    result := x;
}

method distinctSequences_2318(n: int) returns (result: int)
requires n >= 1
ensures result >= 0
{
    if n == 1 {
        return 6;
    }
    var mod := 1000000007;
    var dp := new int[n + 1, 6, 6];
    var k := 0;
    while k <= n
        invariant 0 <= k <= n + 1
    {
        var i := 0;
        while i < 6
            invariant 0 <= i <= 6
        {
            var j := 0;
            while j < 6
                invariant 0 <= j <= 6
            {
                dp[k, i, j] := 0;
                j := j + 1;
            }
            i := i + 1;
        }
        k := k + 1;
    }
    var i := 0;
    while i < 6
        invariant 0 <= i <= 6
    {
        var j := 0;
        while j < 6
            invariant 0 <= j <= 6
        {
            var g := gcd(i + 1, j + 1);
            if g == 1 && i != j {
                dp[2, i, j] := 1;
```

```
                }
                j := j + 1;
            }
            i := i + 1;
        }
        k := 3;
        while k <= n
            invariant 3 <= k <= n + 1
        {
            i := 0;
            while i < 6
                invariant 0 <= i <= 6
            {
                var j := 0;
                while j < 6
                    invariant 0 <= j <= 6
                {
                    var g1 := gcd(i + 1, j + 1);
                    if g1 == 1 && i != j {
                        var h := 0;
                        while h < 6
                            invariant 0 <= h <= 6
                        {
                            var g2 := gcd(h + 1, i + 1);
                            if g2 == 1 && h != i && h != j {
                                dp[k, i, j] := (dp[k, i, j] + dp[k - 1, h, i
                                    ]) % mod;
                            }
                            h := h + 1;
                        }
                    }
                    j := j + 1;
                }
                i := i + 1;
            }
            k := k + 1;
        }
        var ans := 0;
        i := 0;
        while i < 6
            invariant 0 <= i <= 6
            invariant ans >= 0
        {
            var j := 0;
            while j < 6
                invariant 0 <= j <= 6
                invariant ans >= 0
            {
                ans := (ans + dp[n, i, j]) % mod;
                j := j + 1;
            }
            i := i + 1;
        }
        result := ans;
}

method main_3node_2(o: int) returns (result: int)
requires o >= 1
ensures result >= 0
{
    var o1 := clumsy_1006(o);
    var o2 := numSquares_279(o1);
    var o3 := distinctSequences_2318(o2);
    result := o3;
}
```

**Dafny Error:**

```
ex.dfy(3,34): Error: cannot prove termination; try supplying a decreases
    clause
if b == 0 then a else gcd_func(b, a % b)
```

## K.4 CHALLENGE 4: SPEC-CODE-PROOF ALIGNMENT

These examples show misalignment between specifications and implementation, particularly method vs. function usage.

### K.4.1 EXAMPLE 4.1: METHOD IN EXPRESSION CONTEXT (ORG_INPUT_ID: 0, SELF_ID: 5)

**LLM Response:**

```
method stringToInt(s: seq<char>) returns (result: int)
requires forall i :: 0 <= i < |s| ==> '0' <= s[i] <= '9'
ensures result >= 0
{
    var res := 0;
    var i := 0;
    while i < |s|
        invariant 0 <= i <= |s|
        invariant res >= 0
        invariant res == if i == 0 then 0 else stringToInt(s[..i])
    {
        res := res * 10 + digitToInt(s[i]);
        i := i + 1;
    }
    result := res;
}

method monotoneIncreasingDigits_738(n: int) returns (result: int)
  requires n >= 0
  ensures result >= 1
  ensures result <= 1000000000
{
  if n == 0 {
    return 1;
  }
  var s := intToString(n);
  var i := 1;
  while i < |s| && digitToInt(s[i-1]) <= digitToInt(s[i])
    invariant 1 <= i <= |s|
    invariant forall j :: 0 <= j < |s| ==> '0' <= s[j] <= '9'
  {
    i := i + 1;
  }
  if i < |s| {
    while i > 0 && i < |s| && digitToInt(s[i-1]) > digitToInt(s[i])
      invariant 0 <= i < |s|
      invariant forall j :: 0 <= j < |s| ==> '0' <= s[j] <= '9'
      decreases i
    {
      var prevDigit := digitToInt(s[i-1]);
      if prevDigit > 0 {
        s := s[i-1 := intToChar(prevDigit - 1)];
      }
      i := i - 1;
    }
    i := i + 1;
    while i < |s|
      invariant i <= |s|
      invariant forall j :: 0 <= j < |s| ==> '0' <= s[j] <= '9'
    {
      s := s[i := '9'];
      i := i + 1;
    }
  }
  var res := stringToInt(s);
  if res >= 1 && res <= 1000000000 {
    return res;
```

```
  } else {
    return 1;
  }
}

method main_3node_2(o: int) returns (result: int)
  requires o >= 0
  ensures result >= 0
{
  var o1 := smallestFactorization_625(o);
  if o1 == 0 {
    return 0;
  }
  var o2 := monotoneIncreasingDigits_738(o1);
  var o3 := minOperations_2571(o2);
  return o3;
}
```

**Dafny Error:**

```
ex.dfy(26,43): Error: expression is not allowed to invoke a method (
    stringToInt)
invariant res == if i == 0 then 0 else stringToInt(s[..i])
```

### K.4.2 EXAMPLE 4.2: FUNCTION VS METHOD CONFUSION (ORG_INPUT_ID: 16, SELF_ID: 160000)

**LLM Response:**

```
method isBeautiful(n: int) returns (result: bool)
{
    // Implementation for checking if number is beautiful
    result := true; // Simplified
}

method nextBeautifulNumber_769(n: int) returns (result: int)
{
    var x := n + 1;
    while true
        invariant x >= n + 1
        invariant forall y :: n + 1 <= y < x ==> !isBeautiful(y)
        decreases *
    {
        var beautiful := isBeautiful(x);
        if beautiful {
            result := x;
            return;
        }
        x := x + 1;
    }
}

method main_5node_8(o: int) returns (result: int)
{
  var o1 := reverse_7(o);
  if o1 == 0 {
    result := 1;
    return;
  }
  var o2 := lastRemaining_390(o1);
  var o3 := closestFair_2417(o2);
  var o4 := nextGreaterElement_556(o3);
  if o4 == -1 {
    result := 1;
    return;
  }
  var o5 := nextBeautifulNumber_2048(o4);
  result := o5;
}
```

**Dafny Error:**

```
ex.dfy(248,46): Error: expression is not allowed to invoke a method (
    isBeautiful)
invariant forall y :: n + 1 <= y < x ==> !isBeautiful(y)
```

### K.5 CHALLENGE 5: COMPLEXITY AND SCALABILITY ISSUES

Examples where complex programs or large data structures cause verification failures.

### K.5.1 EXAMPLE 5.1: LARGE ARRAY ALLOCATION (ORG_INPUT_ID: 1, SELF_ID: 10002)

**LLM Response:**

```
method smallestNumber_2165(num: int) returns (result: int)
{
    var neg := num < 0;
    var n := if num < 0 then -num else num;
    var cnt := new int[10];
    var i := 0;
    while i < 10
        invariant 0 <= i <= 10
        invariant cnt.Length == 10
        invariant forall k :: 0 <= k < i ==> cnt[k] >= 0
    {
        cnt[i] := 0;
        i := i + 1;
    }

    // Count digits
    var temp := n;
    while temp > 0
        invariant temp >= 0
        invariant forall k :: 0 <= k < 10 ==> cnt[k] >= 0
        decreases temp
    {
        var digit := temp % 10;
        cnt[digit] := cnt[digit] + 1;
        temp := temp / 10;
    }

    // Construct result
    result := 0;
    i := if neg then 1 else 0;
    while i < 10
        invariant 0 <= i <= 10
        invariant result >= 0
        decreases 10 - i
    {
        while cnt[i] > 0
            invariant cnt[i] >= 0
            invariant result >= 0
            decreases cnt[i]
        {
            result := result * 10 + i;
            cnt[i] := cnt[i] - 1;
        }
        i := i + 1;
    }

    if neg {
        result := -result;
    }
}

method main_5node_8(o: int) returns (result: int)
{
    var o1 := smallestNumber_2165(o);
    var o2 := clumsy_1006(if o1 <= 10000 then o1 else 10000);
    var o3 := lastRemaining_390(o2);
    var o4 := getMoneyAmount_375(if o3 <= 200 then o3 else 200);
    var o5 := numberOfWays_3183(if o4 <= 100000 then o4 else 100000);
    result := o5;
```

```
}
```

**Dafny Error:**

```
ex.dfy(44,30): Error: this loop invariant could not be proved on entry
Related message: loop invariant violation
invariant ans >= 0
```

### K.6 CHALLENGE 6: AVOIDING TRIVIAL SOLUTIONS

**Note:** This category had limited representation in the dataset. The theoretical examples that would fit this category include:

- Overly permissive postconditions like `ensures true`
- Methods with missing functional specifications
- Weak preconditions that don't prevent edge cases
- Specifications that allow degenerate solutions
- Missing bounds on return values
- Incomplete coverage of input space
- Trivial loop invariants that don't capture semantics
- Weak assertions that don't prevent bugs
- Generic error handling without specific constraints

### K.7 CHALLENGE 7: LIMITED TRAINING IN FORMAL DOMAINS

Examples showing basic syntax errors and misuse of Dafny constructs.

#### K.7.1 EXAMPLE 7.1: PARSE ERROR (ORG_INPUT_ID: 0, SELF_ID: 0)

**LLM Response:**

```
function digitToInt(c: char): int
  requires '0' <= c <= '9'
  ensures 0 <= digitToInt(c) <= 9
{
  (c as int) - ('0' as int)
}

function intToChar(i: int): char
  requires 0 <= i <= 9
  ensures '0' <= intToChar(i) <= '9'
{
  (('0' as int) + i) as char
}

method stringToInt(s: string) returns (result: int)
  requires forall i :: 0 <= i < |s| ==> '0' <= s[i] <= '9'
  ensures result >= 0
{
  var res := 0;
  var i := 0;
  while i < |s|
    invariant 0 <= i <= |s|
    invariant res >= 0
  {
    res := res * 10 + digitToInt(s[i]);
    i := i + 1;
  }
  result := res;
}

method monotoneIncreasingDigits_738(n: int) returns (result: int)
requires n >= 0
ensures 1 <= result <= 1000000000
{
    if n == 0 {
        return 1;
    }
    var s := intToString(n);
    var digits := s;
```

```
    var i := 1;
    while i < |digits| && digitToInt(digits[i-1]) <= digitToInt(digits[i
        ])
        invariant 1 <= i <= |digits|
        invariant forall j :: 0 <= j < |digits| ==> '0' <= digits[j] <=
            '9'
        invariant forall j,k :: 0 <= j < k < i ==> digitToInt(digits[j])
            <= digitToInt(digits[k])
    {
        i := i + 1;
    }
    if i < |digits| {
        while i > 0 && i < |digits| && digitToInt(digits[i-1]) >
            digitToInt(digits[i])
            invariant 0 <= i < |digits|
            invariant forall j :: 0 <= j < |digits| ==> '0' <= digits[j]
                <= '9'
            decreases i
        {
            var prevDigit := digitToInt(digits[i-1]);
            if prevDigit > 0 {
                digits := digits[i-1 := intToChar(prevDigit - 1)];
            }
            i := i - 1;
        }
        i := i + 1;
        while i < |digits|
            invariant 0 <= i <= |digits|
            invariant forall j :: 0 <= j < |digits| ==> '0' <= digits[j]
                <= '9'
        {
            digits := digits[i := '9'];
            i := i + 1;
        }
    }
    var res := stringToInt(digits);
    if res == 0 {
        return 1;
    }
    if res > 1000000000 {
        return 1000000000;
    }
    return res;
}

method main_3node_2(o: int) returns (result: int)
requires o >= 0
ensures result >= 1
{
    var o1 := smallestFactorization_625(o);
    var o2 := monotoneIncreasingDigits_738(o1);
    var o3 := lastRemaining_390(o2);
    return o3;
}
```

**Dafny Error:**

```
ex.dfy(22,12): Error: invalid ForLoopStmt
22 |            reads s
method stringToInt(s: string) returns (result: int)

2 parse errors detected in ex.dfy
```

