# OpenReview forum: "Local Success Does Not Compose: Benchmarking Large Language Models for Compositional Formal Verification"
_ICLR.cc/2026/Conference — ICLR 2026 Poster_

### Official Review · Reviewer_mpGN · 2025-10-27

**Soundness:** 3
**Presentation:** 3
**Contribution:** 2
**Rating:** 4
**Confidence:** 4

**Summary:**

The paper introduces DAFNYCOMP, a benchmark targeting compositional formal verification for LLMs. It assembles multi-function programs (2–5 functions). Across a series of frontier models, the paper finds a big gap between high syntax success and very low verification results on the benchmark. The authors analyze failures into three primary failure modes: specification fragility, implementation-proof misalignment, and reasoning instability.

**Strengths:**

1. The evaluation is thorough, covering a wide range of models
2. The failure taxonomy is insightful, well-articulated, grounded with examples, and provided with the frequencies.
3. The benchmark is challenging, exposing the limitations of current LLMs.

**Weaknesses:**

1. The benchmark is only done on Dafny. We don't know whether results and claims will still hold for other frameworks such as Verus / Boogie.
2. The synthesis pipeline uses the LeetCode programs. LeetCode skews toward short, single-goal algorithms, but may deviate from the real-world software use cases.
2. There is a limited trial in trying to solve the challenges. E.g., adopting a verifier-in-the-loop repair / agentic setup would be beneficial.

**Questions:**

1. Why do you choose LeetCode problems as the pool programs? Is it possible to switch to other datasets as the initial pools?
2. Will the compositional strategy reflect the distribution of natural real-world programs?
3. How will this methodology translate to other frameworks such as Verus? Does such a similar dataset / idea exist in the Verus or other frameworks' prior work? If not, will your work's pipeline generalize to other frameworks?
4. How to improve the performance of your benchmark? Consider adding a baseline to improve the results

---

> ### Author Response · Authors · 2025-11-21
> **Weakness1 & Weakness2**
>
> We appreciate your thoughtful review and helpful suggestions. We address your points in detail below.
>
> > **Weakness1:** The benchmark is only done on Dafny. We don't know whether results and claims will still hold for other frameworks such as Verus / Boogie.
>
> Thank you for this concern. Comprehensively, our pipeline is also not tied to Dafny: the **Assembl**y step only requires typed functions and call graphs, and the **Formal Translation** step uses template-based LLM prompts plus verifier feedback, which can be instantiated for other SMT-backed frameworks with minor front-end and template changes.
>
> Concretely for **Verus**, the **Assembly** step could more simply operate directly on Rust functions instead of on Python functions, since Verus is built on top of Rust and the existing Rust codebases is substantially larger than that of Dafny. If you want to translate programs from other languages, like Python, into Rust + Verus, our pipeline remains equally applicable, as discussed in our work. The **Translation** step could be re-instantiated with Verus-style templates to ask an LLM to synthesize the final Rust + Verus specifications, using Verus as the checker in the same “AST-based LLM translate + verifier feedback + refinement” loop. Therefore, Implementing a Verus-based version would mainly require these front-end/template changes, not a change to the core pipeline.
>
> Regarding **Boogie**, we would like to clarify that Boogie is already a native part of the Dafny verification pipeline: Dafny programs are first compiled to Boogie, and Boogie then generates the corresponding SMT queries. In this sense, Dafny and Boogie are simply two layers of the same SMT-based verification stack.
>
> And we will add a brief discussion of this comparison to the paper.
>
> ---
>
> > **Weakness2:** The synthesis pipeline uses the LeetCode programs. LeetCode skews toward short, single-goal algorithms, but may deviate from the real-world software use cases.
>
> Thank you for pointing out this limitation. We agree that raw LeetCode problems are biased toward short, single-goal algorithmic solutions and do not, by themselves, reflect the full diversity of real-world software.
>
> Our design tries to **mitigate this limitation** by composing multiple LeetCode-style functions into larger DAG-shaped programs, whose call graphs more closely resemble modular real-world code. In this sense, the synthesized programs are already structurally richer than individual LeetCode solutions, even though they still fall short of full-fledged industrial systems.
>
> Importantly, this limitation stems from our current instantiation rather than from the synthesis method itself. The pipeline is dataset-agnostic and only assumes typed functions with tests; in principle, it can be instantiated with more realistic pools of base functions without modifying the core procedures. We will make this limitation and the corresponding potential extensions explicit in the revised manuscript.

---

> > ### Author Response · Authors · 2025-11-21
> > **Weakness3**
> >
> > > **Weakness3:** There is a limited trial in trying to solve the challenges. E.g., adopting a verifier-in-the-loop repair / agentic setup would be beneficial.
> >
> > We appreciate this suggestion and have conducted further experiments to evaluate an advanced verifier-in-the-loop repair method.
> >
> > Concretely, we set a refinement threshold of **3 turns**: if the model has not resolved the issue within 3 refinement steps, it outputs the final answer from the last turn. At each refinement step, we feed the model both the error message from the verifier and its previous output, asking it to improve its next response. We provide the refinement prompt and the corresponding quantitative results below.
> >
> > ```text
> >  "The Dafny program you produced failed verification. Here is your previous answer:\n"
> > f"```dafny\n{previous_output}\n```\n"
> >  "Dafny verifier responded with:\n"
> > f"```\n{previous_feedback}\n```\n"
> >  "Please fix the Dafny program so that it verifies, keeping the exact signature, requires, and ensures. "
> >  "Return only the Dafny program enclosed in ```dafny``` fences."
> > ````
> >
> > | Model                          | Problems | Pass@turn1 | Syntax@turn1 | Pass@turn2 | Syntax@turn2 | Pass@turn3 | Syntax@turn3 |
> > |--------------------------------|----------|------------|--------------|------------|--------------|------------|--------------|
> > | DeepSeek-R1                    | 300      | 0.33%      | 85.67%       | 5.67%      | 94.33%       | 9.67%      | 95.00%       |
> > | DeepSeek-V3.1                  | 300      | 0.00%      | 49.00%       | 1.67%      | 91.67%       | 5.33%      | 96.00%       |
> > | DeepSeek-V3                    | 300      | 0.00%      | 74.33%       | 2.67%      | 95.00%       | 5.00%      | 96.00%       |
> > | QwQ-32B                        | 300      | 0.00%      | 44.67%       | 0.00%      | 56.67%       | 1.00%      | 61.67%       |
> > | Qwen2.5-Coder-32B-Instruct     | 300      | 0.33%      | 59.67%       | 0.33%      | 62.33%       | 0.67%      | 63.67%       |
> > | Qwen3-Coder-480B-A35B-Instruct | 300      | 0.33%      | 78.33%       | 0.67%      | 93.33%       | 3.00%      | 95.67%       |
> > | gpt-4.1                        | 300      | 0.00%      | 46.67%       | 1.33%      | 86.67%       | 3.33%      | 92.67%       |
> > | gpt-4o                         | 300      | 0.33%      | 94.00%       | 2.00%      | 98.33%       | 4.67%      | 98.67%       |
> > | o4-mini                        | 300      | 0.00%      | 37.33%       | 2.00%      | 92.00%       | 9.67%      | 96.00%       |
> >
> > As shown, we find the multi-turn self-refinement method to be helpful. Although its improvement on the **Syntax Rate** is limited, the **Pass Rate** generally increases once the verifier provides accurate feedback on the errors. At the same time, success rates remain far from saturation, indicating that the benchmark remains challenging even with verifier feedback.
> >
> > Regarding the agentic setups, more elaborate agentic systems are indeed interesting, but systematically exploring their large design space would require substantial effort and is somewhat orthogonal to our current focus on the underlying compositional difficulty of individual models. We therefore regard agentic setups as valuable future work built on top of our benchmark, rather than part of this initial study.
> >
> > Thank you again for the suggestion and we will include the additional evaluation results in the main text.

---

> ### Author Response · Authors · 2025-11-21
> **Question1 & Question2**
>
> > **Question1:** Why do you choose LeetCode problems as the pool programs? Is it possible to switch to other datasets as the initial pools?
>
> We appreciate your concern about using LeetCode as the pool of base programs. Our pipeline is **scalable**, and other Python datasets could also be used as initial pools.
>
> Concretely, the synthesis pipeline is **template-based**: it only assumes a pool of typed functions plus tests. In principle, we could instantiate it with other benchmarks such as HumanEval or MBPP, or with functions mined from real Python repositories, without changing the core procedures.
>
> And in response to the question *“Why do you choose LeetCode problems as the pool programs?”*, we emphasize that **LeetCode** provides a large, publicly available collection of relatively self-contained, single-goal functions that cover a wide range of standard algorithmic patterns. Prior work[1] has shown that **LeetCode** problems exhibit **higher difficulty** and **stronger discriminative power**, which is particularly valuable for constructing meaningful benchmarks. In addition, **LeetCode** offers **reference solutions** and **test cases**. Within our pipeline, we reuse these solutions and tests after composition to efficiently filter out non-compilable or clearly invalid combinations. As a result, LeetCode serves as a compact yet structurally rich source of typed building blocks with ground-truth behavior, making it an ideal choice for our initial pools.
>
> [1] LeetCodeDataset: A Temporal Dataset for Robust Evaluation and Efficient Training of Code LLMs
> [https://arxiv.org/abs/2504.14655v1](https://arxiv.org/abs/2504.14655v1)
>
> ---
>
> > **Question2:** Will the compositional strategy reflect the distribution of natural real-world programs?
>
> We acknowledge the importance of aligning with the distribution of real-world programs. To this end, we **extend** our benchmark with 100 additional programs featuring non-chain DAG structures. Specifically, we generate 10 instances for each of 10 distinct topology types, all constructed through the same pipeline. This augmentation **broadens the structural diversity** of the benchmark while preserving pipeline consistency.
>
> | ID | Topology Description |
> |----|-----------------------|
> | 1 | 4 nodes, root splits into two, one branch splits again |
> | 2 | 4 nodes, root splits into three |
> | 3 | 4 nodes, chain with a branch at the second node |
> | 4 | 5 nodes, chain then splits into three |
> | 5 | 5 nodes, root splits into four |
> | 6 | 5 nodes, root splits into three, leftmost branch splits again |
> | 7 | 5 nodes, chain with a branch at the second node, both branches split again |
> | 8 | 5 nodes, root splits into two, left branch splits again |
> | 9 | 5 nodes, root splits into two, both branches are chains of two |
> | 10 | 5 nodes, root splits into two, one branch is a chain of three |
>
> And to be specific, our compositional strategy is explicitly designed to approximate key structural aspects of such programs rather than isolated LeetCode-style solutions. We begin from single-responsibility functions and compose them into larger call graphs, propagating and reusing intermediate values. In this way, the resulting programs exhibit multiple interacting subroutines and cross-call dependencies, patterns deliberately designed to approximate the modular organization of real-world code.
>
> Therefore we view it as **distributionally closer but still simplified**: it reflects realistic compositional and dependency patterns at the function-call level, while remaining small enough to analyze systematically. It can be further instantiated with components mined from real-world code in future work.

---

> ### Author Response · Authors · 2025-11-21
> **Question3 & Question4**
>
> > **Question3:** How will this methodology translate to other frameworks such as Verus? Does such a similar dataset / idea exist in the Verus or other frameworks' prior work? If not, will your work's pipeline generalize to other frameworks?
>
> Thank you for pointing it out. Comprehensively, our pipeline is designed to be scalable and can translate to other **ATP (Automated Theorem Proving)** frameworks like **Dafny/Verus/Boogie**.
>
> Concretely for **Verus**, the **Assembly** step could more simply operate directly on Rust functions instead of on Python functions, since Verus is built on top of Rust and the existing Rust codebases is substantially larger than that of Dafny. If you want to translate programs from other languages, like Python, into Rust + Verus, our pipeline remains equally applicable, as discussed in our work. The **Translation** step could be re-instantiated with Verus-style templates to ask an LLM to synthesize the final Rust + Verus specifications, using Verus as the checker in the same “AST-based LLM translate + verifier feedback + refinement” loop. Therefore, Implementing a Verus-based version would mainly require these front-end/template changes, not a change to the core pipeline.
>
> In response to the question *“Does such a similar dataset / idea exist in the Verus or other frameworks’ prior work?”,* we note that there are two closely related lines of work in the Verus ecosystem. An important Verus benchmark, **Verus-Bench**, is provided by **AutoVerus [1]**, but its 150 proof tasks are **manually curated** and annotated from existing benchmark suites. By contrast, **SAFE [2]** adopts a self-evolving pipeline with iterative data synthesis and **fine-tuning** to learn a dedicated specification generator, and thereby constructs the dataset **CodeNet-Test**. In particular, whether the functions are manually curated or generated via fine-tuned models, they almost never exhibit a meaningful call graph, which is a key ingredient for assessing a model’s compositional reasoning. Therefore, Our contribution is complementary: we systematically assemble compositional programs to stress-test specification tracking through call graphs. We will add a brief discussion of this comparison to the paper.
>
> [1] AutoVerus: Automated Proof Generation for Rust Code.
> [https://arxiv.org/abs/2409.13082](https://arxiv.org/abs/2409.13082)
>
> [2] SAFE: Automated Proof Generation for Rust Code via Self-Evolution.
> [https://openreview.net/forum?id=2NqssmiXLu](https://openreview.net/forum?id=2NqssmiXLu)
>
> ---
>
> > **Question4:** How to improve the performance of your benchmark? Consider adding a baseline to improve the results.
>
> Thank you for this insightful suggestion. Based on your valuable feedback in **Weakness 3**, we have added a multi-turn repair baseline to improve performance on our benchmark, following the verifier-in-the-loop repair procedure you suggested.
> | Model                          | Problems | Pass@turn1 | Syntax@turn1 | Pass@turn2 | Syntax@turn2 | Pass@turn3 | Syntax@turn3 |
> |--------------------------------|----------|------------|--------------|------------|--------------|------------|--------------|
> | DeepSeek-R1                    | 300      | 0.33%      | 85.67%       | 5.67%      | 94.33%       | 9.67%      | 95.00%       |
> | DeepSeek-V3.1                  | 300      | 0.00%      | 49.00%       | 1.67%      | 91.67%       | 5.33%      | 96.00%       |
> | DeepSeek-V3                    | 300      | 0.00%      | 74.33%       | 2.67%      | 95.00%       | 5.00%      | 96.00%       |
> | QwQ-32B                        | 300      | 0.00%      | 44.67%       | 0.00%      | 56.67%       | 1.00%      | 61.67%       |
> | Qwen2.5-Coder-32B-Instruct     | 300      | 0.33%      | 59.67%       | 0.33%      | 62.33%       | 0.67%      | 63.67%       |
> | Qwen3-Coder-480B-A35B-Instruct | 300      | 0.33%      | 78.33%       | 0.67%      | 93.33%       | 3.00%      | 95.67%       |
> | gpt-4.1                        | 300      | 0.00%      | 46.67%       | 1.33%      | 86.67%       | 3.33%      | 92.67%       |
> | gpt-4o                         | 300      | 0.33%      | 94.00%       | 2.00%      | 98.33%       | 4.67%      | 98.67%       |
> | o4-mini                        | 300      | 0.00%      | 37.33%       | 2.00%      | 92.00%       | 9.67%      | 96.00%       |
>
> The multi-turn verifier-in-the-loop repair already yields non-trivial gains over the single-shot or 8-shot results, but success rates remain far from saturation for all models, indicating that the benchmark continues to pose a substantial challenge even with verifier feedback.
>
> At the same time, although more advanced agentic systems are appealing directions, fully exploring their large design space would be costly and is somewhat orthogonal to our current emphasis on underlying compositional difficulty of individual models. And we therefore view agentic systems as valuable future baselines built on top of our benchmark, rather than part of this initial study.

---

### Official Review · Reviewer_2F9W · 2025-10-29

**Soundness:** 2
**Presentation:** 2
**Contribution:** 2
**Rating:** 2
**Confidence:** 4

**Summary:**

This paper presents a benchmark set, DafnyComp, to evaluate LLMs ability to perform compositional
reasoning. The benchmark set contains Dafny programs that are compositions of 2-5 functions. The goal
is to prove these programs correct. To do this, one has to generate pre and postconditions (requires
and ensures) for each function in a way that they compose to yield a proof for the main program.
The LLMs task is to annotate the functions enough to enable Dafny verifier to complete the proof
(by discharging the verification conditions using an SMT solver).

The main result shows that LLMs across the board are really bad at doing this annotation task.
They are able to do so for single functions to some extent, but really struggle to do that for
multiple functions in a way that the annotations compose.

**Strengths:**

Strengths:
1. DafnyComp is a valuable benchmark since state-of-the-art LLMs currently perform very poorly
(about 3% for the best models).
2. DafnyComp tests the ability to think compositionally, which is useful to evalaute.
3. DafnyComp is also interesting because models can generate correct syntax (for annotations) with
high probability, so it does demonstrate that models really struggle with compositional reasoning
(and not getting the syntax right for a potentially low-resource language).
4. The pipeline for constructing DafnyComp guarantees that the benchmarks are all high quality
(i.e. they all can be formally proved correct.)

**Weaknesses:**

Weaknesses:

1. The benchmarks in DafnyComp are artificial - the individual functions are natural, but the
compositions are contrived (authors, please correct me if I am wrong.)  So, the 92% drop from
single-function to multi-function programs can also be attributed to the fact that the single-function
programs are natural programs that people may write and care for, and the multi-function programs
are more arbitrary. In other words, if I were to write completely unnatural single function programs,
wouldn't LLMs performance already deteriorate? That puts the main claim into question: maybe it is
not the lack of compositional reasoning, but just that the compositional programs are unnatural.
If the single-function programs were unnatural, or the compositional programs were natural, then
would we get different conclusions?

2. The paper makes very strongly worded claims all through the paper, but there are a lot of
factors that could be responsible for the observed behavior. Many such factors are overlooked and
not discussed.

 2.1 If the function call graph is linear (which means it can be topologically sorted), then couldn't
we turn the multi-function benchmarks to single-function benchmarks? If so, then would LLMs
perform better on these derived single-function benchmarks? And would this finding impact the main
claims of the paper?

 2.2 Is Dafny prover guaranteed to prove things if they are provable -- couldn't the SMT solver fail --
e.g. time out -- even when the verification condition holds true? Would this not somehow impact
the conclusions? For example, purely hypothetically, maybe the LLM is generating the correct
annotations and the SMT solver is failing to prove them? I know that this is unlikely, but how
are we guarding against that possibility?

3. There is not enough explanation for how benchmarks (and their correctness assertions) are
generated.

 3.1 It is unclear how the assertions of the composed program are generated. Line 215 says "We execute
each composed program against the reference unit tests from LeetCodeDataset" -- but my understanding
is that you are taking individual functions from LeetCodeDataset and then creating new compositions
(with 2-5 functions). So, LeetCodeDataset would not have reference unit tests for these composed
programs. There is clearly some misunderstanding here on my part.

 3.2 Some of these misunderstandings could be avoided if you had a running example either in the main
paper or at least in the Appendix. It would be good to have an example benchmark in DafnyComp. My best
Source is Appendix G. But most of the programs there have 1 function -- clearly, they can't be part
of DafnyComp. So, that adds to more confusion. G.4.2 is the first place where there is a program that
uses multiple functions. It has one function, reverse_7, that is unused, that is a but confusing, but
maybe DafnyComp has benchmarks that contain unused functions. But what is really confusing is that
"nextBeautifulNumber_769" is not a linear composition of two functions, but a loop over the "isBeautiful"
function.

 3.3 The examples in Appendix G just give some code snippet and a Dafny error. That is not helpful.
More explanation is needed. I could not understand most examples. For example, take G.1.1: sum >= 0
is the only postcondition there. It is clearly implied by the loop invariant (because it contains
a conjunct sum >= 0) -- but Dafny error says "a postcondition could not be proved..." ?

Other questions:

- Contribution (ii) on Line 111 is related to a single function/procedure, right? So, it is not
related to compositions, is that true?

- Does DafnyComp have 300 benchmarks or 900 benchmarks (Table 2)? Or Table 2 is analyzing 3 runs for
each of the 300 benchmarks?

- Line 422-423: "not a typo but a systematic failure to..." -- what do you mean by "not a typo"?

- Line 427-428: "attention mechanisms excel at ...implementations." I think there is no experiment
in the paper that provides evidence of this claim, and I am not even sure what this claim means.

**Questions:**

Please refer to the Weaknesses section for the questions.

---

> ### Author Response · Authors · 2025-11-21
> **Weakness1 & Weakness2.1(1)**
>
> We greatly appreciate your detailed feedback and thoughtful remarks. We address each of your comments in turn below.
>
> > **Weakness1:** The benchmarks in DafnyComp are artificial … If the single-function programs were unnatural, or the compositional programs were natural, then would we get different conclusions?
> >
> > **Weakness2.1:** If the function call graph is linear (which means it can be topologically sorted), then couldn't we turn the multi-function benchmarks to single-function benchmarks? If so, then would LLMs perform better on these derived single-function benchmarks? And would this finding impact the main claims of the paper?
>
> Thank you for raising these concerns. And we address these two closely related questions jointly, since they can be answered by a single extended experiment.
>
> Comprehensively, we find that after **transforming each composed program into an equivalent single-function** implementation, only the **Syntax Rate** changes marginally, while our main claim on the **Verified Rate** remains unaffected.
>
> Concretely, to estimate artificial style bias, we first **transform each composed program into an equivalent single-function implementation.** A template illustrating this procedure is provided below:
>
> ```text
> Func A: BlockA; Func B: BlockB; Func C: BlockC; main_control: A→B→C
>
> ⇒
>
> Func A_B_C: BlockA → BlockB → BlockC
> ````
>
> We then rerun our evaluation under this single-function setting and manually inspect 10 representative failure cases from the initial 8-shot evaluation. We conduct the evaluation under both the standard 8-shot approach and a stronger multi-turn refinement paradigm:
>
> | Setting    | Model                           | Syntax Corret | Verified | Failure |
> |------------|---------------------------------|---------------|----------|---------|
> | Compose    | Pro/deepseek-ai/DeepSeek-V3     | 1             | 1        | 9 / 10  |
> | multi-turn | Pro/deepseek-ai/DeepSeek-V3.1   | 0             | 1        | 9 / 10  |
> | Single     | Pro/deepseek-ai/DeepSeek-V3     | 0             | 1        | 9 / 10  |
> | multi-turn | Pro/deepseek-ai/DeepSeek-V3.1   | 0             | 1        | 9 / 10  |
> | Compose    | Pro/deepseek-ai/DeepSeek-V3     | 1             | 0        | 10 / 10 |
> | 8-shot     | Pro/deepseek-ai/DeepSeek-V3.1   | 0             | 0        | 10 / 10 |
> | Single     | Pro/deepseek-ai/DeepSeek-V3     | 0             | 0        | 10 / 10 |
> | 8-shot     | Pro/deepseek-ai/DeepSeek-V3.1   | 0             | 0        | 10 / 10 |
>
> After this transformation, the main difference we observe is in **syntactic correctness**: some models produce slightly more syntactically well-formed programs on single functions. But **this does not materially affect the final verified rate**. As shown in **Table 1** of our paper, models like **Claude-3.5-Sonnet-20241022** can have a **lower Syntax Rate (90.67% < 95.67%, 96.33% < 97.33%)** than **Claude-4-Sonnet-20250514**, while still achieving a **higher Verified Rate (3.67% > 2.33%, 4.67% > 3.00).** We observe the same trend in the full multi-turn refinement results as below:
>
> | Model                          | Problems | Pass@turn1 | Syntax@turn1 | Pass@turn2 | Syntax@turn2 | Pass@turn3 | Syntax@turn3 |
> |--------------------------------|----------|------------|--------------|------------|--------------|------------|--------------|
> | DeepSeek-R1                    | 300      | 0.33%      | 85.67%       | 5.67%      | 94.33%       | 9.67%      | 95.00%       |
> | DeepSeek-V3.1                  | 300      | 0.00%      | 49.00%       | 1.67%      | 91.67%       | 5.33%      | 96.00%       |
> | DeepSeek-V3                    | 300      | 0.00%      | 74.33%       | 2.67%      | 95.00%       | 5.00%      | 96.00%       |
> | QwQ-32B                        | 300      | 0.00%      | 44.67%       | 0.00%      | 56.67%       | 1.00%      | 61.67%       |
> | Qwen2.5-Coder-32B-Instruct     | 300      | 0.33%      | 59.67%       | 0.33%      | 62.33%       | 0.67%      | 63.67%       |
> | Qwen3-Coder-480B-A35B-Instruct | 300      | 0.33%      | 78.33%       | 0.67%      | 93.33%       | 3.00%      | 95.67%       |
> | gpt-4.1                        | 300      | 0.00%      | 46.67%       | 1.33%      | 86.67%       | 3.33%      | 92.67%       |
> | gpt-4o                         | 300      | 0.33%      | 94.00%       | 2.00%      | 98.33%       | 4.67%      | 98.67%       |
> | o4-mini                        | 300      | 0.00%      | 37.33%       | 2.00%      | 92.00%       | 9.67%      | 96.00%       |
>
> This suggests that, while formatting and style can shift the syntactic success rate, they do **not** drive our main conclusions about compositional reasoning. What matters for our study is the **verified rate**—i.e., the model’s ability to synthesize code that satisfies all constraints of the call graph and passes the Dafny verifier. Under this metric, our findings remain stable even after converting compositions into single functions.

---

> ### Author Response · Authors · 2025-11-21
> **Weakness1 & Weakness2.1(2)**
>
> Additionally, we respectfully disagree that our composed setting is merely “artificial.” Our construction is analogous in spirit to benchmarks such as **ARC-AGI[1,2]**, where simple, familiar modules are **intentionally recombined under human-designed rules**, encouraging models to reason about relationships between components rather than memorize style patterns.
>
> In our case, we compose individually meaningful functions into larger programs while preserving standard modular coding practices. The logical constraints are not arbitrary: they are the **natural contracts** attached to each component, distributed across the call graph. The goal is precisely to test whether a model can integrate these scattered constraints and perform **compositional reasoning**, where developers routinely combine library functions and internal utilities.
>
> Our benchmark should therefore be seen as a controlled but **realistically motivated** probe of this ability: we do not change the semantics of the original components, but only challenge the model to maintain global correctness when they are combined into more complex topologies.
>
> [1] On the Measure of Intelligence.
> [https://arxiv.org/abs/1911.01547](https://arxiv.org/abs/1911.01547)
>
> [2] ARC-AGI-2: A New Challenge for Frontier AI Reasoning Systems.
> [https://arxiv.org/abs/2505.11831](https://arxiv.org/abs/2505.11831)

---

> ### Author Response · Authors · 2025-11-21
> **Weakness2.2**
>
> > **Weakness2.2:** Is Dafny prover guaranteed to prove things if they are provable -- couldn't the SMT solver fail -- e.g. time out -- even when the verification condition holds true? Would this not somehow impact the conclusions? For example, purely hypothetically, maybe the LLM is generating the correct annotations and the SMT solver is failing to prove them? I know that this is unlikely, but how are we guarding against that possibility?
>
> We agree that Dafny’s prover is not complete: even when a verification condition is valid, the underlying SMT solver may in principle fail to prove it within a given resource budget. We conduct additional experiments to guard against the possibility of verification errors by the verifier.
>
> Concretely, in our original evaluation, if the Dafny verifier does not complete within **60 seconds**, we treat the instance as a failure with status **execute_error**. We chose 60 seconds because it is a relatively generous budget for verifying Dafny code; if the solver still times out under this budget, this typically indicates that the verification condition derived from the whole program and its specification lies beyond what the SMT backend can practically handle. This suggests both that (i) the model has not produced a sufficiently reasonable program–specification pair, and (ii) simply increasing the timeout is unlikely to convert such cases into successful proofs. For completeness, we conduct additional experiments that re-run the originally timed-out samples with a larger timeout and report the resulting verification outcomes below. As an illustrative case, we take the integer-valued timeout samples from **Qwen3-Coder-480B-A35B-Instruct** as an example.
>
> | Setting          | Value                          |
> |------------------|--------------------------------|
> | Model            | Qwen3-Coder-480B-A35B-Instruct |
> | #Samples         | 60           ,  60,  60        |
> | Timeout limit(s) | 60 (original), 120, 300        |
>
> As shown, increasing the timeout from 60 s to 120 s and 300 s does not change the outcome: all 60 samples remain unverified. This suggests that, under our timeout settings, prover incompleteness has limited practical impact on our conclusions, and our results remain stable under larger time budgets.

---

> ### Author Response · Authors · 2025-11-21
> **Weakness3.1**
>
> > **Weakness3.1:** It is unclear how the assertions of the composed program are generated. Line 215 says "We execute each composed program against the reference unit tests from LeetCodeDataset" -- but my understanding is that you are taking individual functions from LeetCodeDataset and then creating new compositions (with 2-5 functions). So, LeetCodeDataset would not have reference unit tests for these composed programs. There is clearly some misunderstanding here on my part.
>
> Thanks for pointing out this confusion. Instead of directly using the unit tests provided in the LeetCodeDataset, we reconstruct unit tests based on the call graph of the composed functions and their logical constraints.
>
> Concretely, what we actually do is to **derive** unit tests for each composed function from the unit tests of its constituent LeetCode functions. Each individual LeetCode function comes with input–output examples, which we view as admissible input–output constraints on that function. When we construct a composed function from 2–5 such functions, we propagate these admissible constraints through the composition and intersect them: we require that there exist intermediate values that simultaneously satisfy the admissible input–output restrictions of all constituent functions along the composition. The intersection of these propagated constraints defines the valid unit tests for the composed function. If this intersection turns out to be empty—i.e., no consistent test cases can be constructed—verification becomes impossible, and we simply discard that candidate sample.
>
> Thus, the “reference unit tests from LeetCodeDataset’’ mentioned on L215 are **not** pre-existing tests for the composed programs, but instead the induced unit tests obtained by intersecting the input–output constraints of the underlying LeetCode functions. In the revised manuscript, we make this explicit by adding the following sentence: *“…to confirm functional correctness and basic executability. We construct unit tests for composed functions by intersecting the input–output constraints of their constituent functions. If the intersection is empty, the sample is discarded.”*

---

> ### Author Response · Authors · 2025-11-21
> **Weakness3.2 & Weakness3.3**
>
> > **Weakness3.2:** Some of these misunderstandings could be avoided if you had a running example either in the main paper or at least in the Appendix. It would be good to have an example benchmark in DafnyComp. My best Source is Appendix G. But most of the programs there have 1 function -- clearly, they can't be part of DafnyComp. So, that adds to more confusion. G.4.2 is the first place where there is a program that uses multiple functions. It has one function, reverse_7, that is unused, that is a but confusing, but maybe DafnyComp has benchmarks that contain unused functions. But what is really confusing is that "nextBeautifulNumber_769" is not a linear composition of two functions, but a loop over the "isBeautiful" function.
>
> We acknowledge that the current presentation of **Appendix G** may have caused some confusion, because it only shows the failing code snippets rather than the full programs.
>
> **Appendix G** presents **only failing code sections** from larger multi-function programs. In doing so, we displayed only the error location and omitted the broader compositional context. The inclusion of **reverse_7** in **G.4.2** was unintended. In addition, **isBeautiful** in **G.4.2** is a helper function **automatically** generated by the LLM during Dafny synthesis. This is standard practice in formal verification, where human programmers also introduce auxiliary functions to modularize verification. The compositional challenge remains: loop invariants must reference **isBeautiful**’s specification.
>
> We have revised **Appendix G** to show full method bodies instead of isolated snippets, thereby making the examples more self-contained and easier to understand.
>
> ---
>
> > **Weakness3.3:** The examples in Appendix G just give some code snippet and a Dafny error. That is not helpful. More explanation is needed. I could not understand most examples. For example, take G.1.1: sum >= 0 is the only postcondition there. It is clearly implied by the loop invariant (because it contains a conjunct sum >= 0) -- but Dafny error says "a postcondition could not be proved..." ?
>
> We appreciate the reviewer’s feedback on Appendix G and agree that the current presentation with brief code snippets and Dafny error messages is not sufficiently self-contained.
>
> In response to the issue of **“G.1.1”**, we note that we only showed a snippet of the failing code without the full context. In the actual benchmark, **digitSum** is called by another function **largestPalindrome** (which we did not include). The ground truth specification has two postconditions: **ensures sum >= 0** *and* **ensures sum == DigitSumSpec(n)**. The LLM only generated the first one. While this weaker postcondition is locally provable, it is compositionally insufficient—the downstream caller needs to know the relationship between **sum** and **n** to verify its own logic, not just that **sum** is non-negative.
>
> We have revised **Appendix G** to show full method bodies instead of isolated snippets, thereby making the examples more self-contained and easier to understand.

---

> ### Author Response · Authors · 2025-11-21
> **Other Questions**
>
> > **Question1:** Contribution (ii) on Line 111 is related to a single function/procedure, right? So, it is not related to compositions, is that true?
>
> Thank you for pointing this out. While we understand the perspective, we believe this characterization may not fully reflect our intent.
>
> Concretely, misalignment between implementation and proof is typically detected at the level of individual functions or procedures, yet it naturally becomes an obstacle for compositional reasoning as it undermines the ability to reliably combine and reuse verified components: if any callee’s implementation and specification are misaligned, the entire composed proof can fail even when other components are locally reasonable.
>
> In the revised manuscript, we clarify this connection by extending the description of Contribution (ii) as follows: “*…any of which cause verification to fail despite syntactic well-formedness, and in composed programs they immediately block end-to-end proofs because component contracts can no longer be trusted as summaries.*”
>
> ---
>
> > **Question2:** Does DafnyComp have 300 benchmarks or 900 benchmarks (Table 2)? Or Table 2 is analyzing 3 runs for each of the 300 benchmarks?
>
> Thank you for catching this ambiguity. We clarify this in the revised **Table 2** caption as: "*Distribution of verification failure modes across 900 analyzed cases (3 best-performing models × 300 benchmarks) from DafnyComp.*”
>
> Concretely, the 900 cases come from evaluating the top 3 performing models (**Claude-4-Sonnet**, **Deepseek-R1**, and **Gemini-2.5-Pro**) on all 300 benchmarks: 3 models × 300 benchmarks = 900 test cases. We chose these three models because they achieved the highest verification rates and provided the most diverse failure patterns for analysis. This sampling strategy allows us to identify systematic failure modes across different model architectures while keeping the manual analysis tractable.
>
> ---
>
> > **Question3:** Line 422-423: "not a typo but a systematic failure to..." -- what do you mean by "not a typo"?
>
> Thank you for pointing out this ambiguity. Our intention was not to suggest a literal character-level typo by the model, but to emphasize that such contradictory assertions are **not isolated one-off accidents**: they appear as part of a broader pattern, contributing to the 21.7% of failures where syntactically valid code contradicts its own specifications.
>
> To avoid this ambiguity, in the revised manuscript, we rephrase the sentence as follows: *“…we observe assertions such as assert 0 >= 1; inside otherwise reasonable code, illustrating a recurring pattern of internally contradictory assertions rather than an isolated slip.”*
>
> ---
>
> > **Question4:** Line 427-428: "attention mechanisms excel at ...implementations." I think there is no experiment in the paper that provides evidence of this claim, and I am not even sure what this claim means.
>
> Thanks for pointing this out. We want to clarify that our intent was only to relate our observations to existing mechanistic analyses of transformer circuits, not to make a new architecture-level claim.
>
> To address this concern, we have revised the discussion in **Section 5.3** to avoid over-interpretation. In the updated version, we replace the previous, stronger wording with the following more cautious sentence: *“This pattern is consistent with mechanistic analyses of transformer attention heads, which typically reveal relatively local algorithmic behaviors rather than globally consistent reasoning[1][2].”* We believe this phrasing better reflects the scope of our evidence and clearly positions our remark as a connection to, rather than a consequence of, existing mechanistic studies.
>
> [1] A Mathematical Framework for Transformer Circuits: [https://transformer-circuits.pub/2021/framework/index.html](https://transformer-circuits.pub/2021/framework/index.html)
>
> [2] In-context Learning and Induction Heads: [https://transformer-circuits.pub/2022/in-context-learning-and-induction-heads/index.html](https://transformer-circuits.pub/2022/in-context-learning-and-induction-heads/index.html)

---

### Official Review · Reviewer_KdjF · 2025-11-03

**Soundness:** 3
**Presentation:** 3
**Contribution:** 4
**Rating:** 8
**Confidence:** 4

**Summary:**

The paper introduces DafnyComp, a benchmark to evalaute LLMs in . Unlike existing benchmarks, DafnyComp focuses on programs consisting of multiple functions interacting with each other. Experiments on 13 frontier LLMs show that this leads to significantly a more challenging and realistic benchmark.

**Strengths:**

* The paper introduces DafnyComp, a benchmark for evaluating LLMs on the generation of compositional specifications in Dafny. Existing benchmarks for verifiable code generation only test problems with a single function. DafnyComp focuses on programs composed of multiple interacting functions with necessary data dependencies, addressing a significant gap in the real-world applications of formal verification.
* A comprehensive set of LLMs is evaluated on DafnyComp. Results show that multi-function compositional problems present a major challenge to LLMs compared to simpler problems.
* The authors performed a detailed and insightful analysis of LLMs' failures on DafnyComp.

**Weaknesses:**

* The benchmark is relatively small (300 programs). Is it possible to scale the benchmark construction method in Sec. 3 to collect more programs?
* The paper focuses on Dafny. It would be great to discuss potential extensions to other systems such as Lean and Verus.

**Questions:**

N/A

---

> ### Author Response · Authors · 2025-11-21
> **Weakness1 & Weakness2**
>
> We thank you for your positive and thoughtful review, and for the constructive suggestions. We address your remaining comments point by point below.
>
> > **Weakness1:** The benchmark is relatively small (300 programs). Is it possible to scale the benchmark construction method in Sec. 3 to collect more programs?
>
> We thank you for raising this concern and for specifically inquiring about the scalability of our construction method. Comprehensively, we want to clarify that our composition pipeline is **template-based** rather than hard-coded, which could be naturally scaled to generate diverse DAG-style structures.
>
> Additionally, we generate 100 new programs featuring **non-chain DAG structures** (10 instances of each of the following 10 topology types):
>
> | ID | Topology Description |
> |----|-----------------------|
> | 1 | 4 nodes, root splits into two, one branch splits again |
> | 2 | 4 nodes, root splits into three |
> | 3 | 4 nodes, chain with a branch at the second node |
> | 4 | 5 nodes, chain then splits into three |
> | 5 | 5 nodes, root splits into four |
> | 6 | 5 nodes, root splits into three, leftmost branch splits again |
> | 7 | 5 nodes, chain with a branch at the second node, both branches split again |
> | 8 | 5 nodes, root splits into two, left branch splits again |
> | 9 | 5 nodes, root splits into two, both branches are chains of two |
> | 10 | 5 nodes, root splits into two, one branch is a chain of three |
>
> Therefore, scaling our pipeline only requires defining and adding new **topology-structure templates** and adjusting the **prompts** as needed for the target tasks. **No other modifications are required to the underlying assembly or formal-translation procedures.**
>
> ---
>
> > **Weakness2:** The paper focuses on Dafny. It would be great to discuss potential extensions to other systems such as Lean and Verus.
>
> Thank you for point it out. Comprehensively, our pipeline is designed to be scalable and can translate to other **ATP (Automated Theorem Proving)** frameworks like **Dafny/Verus/Boogie**.
>
> Concretely for **Verus**, the **Assembly** step could more simply operate directly on Rust functions instead of on Python functions, since Verus is built on top of Rust and the existing Rust codebases is substantially larger than that of Dafny. If you want to translate programs from other languages, like Python, into Rust + Verus, our pipeline remains equally applicable, as discussed in our work. The **Translation** step could be re-instantiated with Verus-style templates to ask an LLM to synthesize the final Rust + Verus specifications, using Verus as the checker in the same “AST-based LLM translate + verifier feedback + refinement” loop. Therefore, Implementing a Verus-based version would mainly require these front-end/template changes, not a change to the core pipeline.
>
> Regarding **Lean**, we agree that exploring such extensions would be interesting but unlike Dafny, Lean is based on **ITP (Interactive Theorem Proving)**. Consequently, the workflows, user interaction patterns, and proof obligations differ substantially from the setting studied in this paper. Since we have not yet conducted systematic experiments or design work in **Lean**, we prefer not to speculate about concrete extensions. Instead, we view this as an intriguing direction for future work and will add a brief discussion of this comparison to the paper.

---

> > ### Comment · Reviewer_KdjF · 2025-11-25
> >
> > Thanks for the clarification. I'll maintain my current score.

---

> > > ### Author Response · Authors · 2025-11-26
> > >
> > > We sincerely thank you for your thoughtful feedback and for taking the time to carefully review our responses. Thank you for helping us improve our paper.

---

### Official Review · Reviewer_M3WA · 2025-11-04

**Soundness:** 2
**Presentation:** 2
**Contribution:** 2
**Rating:** 4
**Confidence:** 3

**Summary:**

The paper presents DAFNYCOMP, a collection of 300 programs used to provide stress on the compositional generation of specifications in Dafny by chaining 2-5 functions with real dependency of data. In 13 frontier LLMs, syntax is well-behaved (96%-99% Pass at t8) but the verification is still low (3.7 percent overall; best model about 7 percent), indicating that there is a big discrepancy between syntax and semantic correctness. The failures the authors consider are three in general, namely specification fragility, implementation-proof misalignment, and reasoning instability.

**Strengths:**

- Timely evaluation of compositional verification
- Clear, reproducible benchmark spec with Dafny ground truth.
- Concrete numbers that expose a large syntax verification gap.
- Insightful error analysis

**Weaknesses:**

- Data programs and specs are generated using an LLM from a single family employed in evaluation, which might be biased toward one style and negatively impact others.
- The evaluation is limited to topologies: only chains are considered, whereas real systems include trees, DAGs, shared states, and more complex dependencies.
- There is a lack of evaluation on more advanced test-time scaling strategies, such as multi-turn self-refinement.
- The evaluation described in the main text does not consistently standardize or control details about compute and prompting across models (tokens, temperature, verifier retry policy).

**Questions:**

See weaknesses.

---

> ### Author Response · Authors · 2025-11-21
> **Weakness1**
>
> We appreciate your engagement with our work and your valuable insights. We carefully address the points you raised as follows.
>
> > **Weakness1:** Data programs and specs are generated using an LLM from a single family employed in evaluation, which might be biased toward one style and negatively impact others.
>
> Thank you for pointing this out. Comprehensively, we choose **Claude-4-Sonnet-20250514** as our generation model because it performs **the best** within our formal translation pipeline. And we include details on model selection for formal translation below, which leads to our choice of **Claude-4-Sonnet-20250514**.
>
> Concretely, we randomly sampled **100** test cases from the **564** instances obtained after our assembly procedure to evaluate each model with identical prompts and decoding settings. We define **success rate** as the fraction of generations whose specifications are valid under the Dafny verifier. We have now added them for reference as **Table 4 in Appendix D (”Model Selection for Formal Translation”)**.
>
> | Model              | Success Ratio |
> |--------------------|---------------|
> | Claude 4 Sonnet    | 58.00%        |
> | Claude 3.5 Sonnet  | 55.00%        |
> | gpt-3.5-turbo      | 45.00%        |
> | gpt-4o             | 31.00%        |
> | gpt-4o-mini        | 41.00%        |
> | o1                 | 36.00%        |
> | o1-mini            | 33.00%        |
> | o3-mini            | 37.00%        |
> | gemini-2.0-flash   | 38.00%        |
>
> As shown, we ultimately chose **Claude-4-Sonnet-20250514** because it yielded the highest rate of valid specifications on these cases. This aligns with our design goal to **prioritize the quality of generated specifications** used downstream in our experiments.
>
> Additionally, to further ensure fairness in the corresponding evaluation, we will remove the Claude results and revise the paragraph accordingly in the updated version.

---

> ### Author Response · Authors · 2025-11-21
> **Weakness2**
>
> > **Weakness2:** The evaluation is limited to topologies: only chains are considered, whereas real systems include trees, DAGs, shared states, and more complex dependencies.
>
> Thanks for this great suggestion! We include additional design tasks involving other DAG-style compositions and generated **100** new programs featuring **non-chain DAG structures** (10 instances of each of the following 10 topology types):
>
> | ID | Topology Description |
> |----|-----------------------|
> | 1 | 4 nodes, root splits into two, one branch splits again |
> | 2 | 4 nodes, root splits into three |
> | 3 | 4 nodes, chain with a branch at the second node |
> | 4 | 5 nodes, chain then splits into three |
> | 5 | 5 nodes, root splits into four |
> | 6 | 5 nodes, root splits into three, leftmost branch splits again |
> | 7 | 5 nodes, chain with a branch at the second node, both branches split again |
> | 8 | 5 nodes, root splits into two, left branch splits again |
> | 9 | 5 nodes, root splits into two, both branches are chains of two |
> | 10 | 5 nodes, root splits into two, one branch is a chain of three |
>
> Corresponding evaluation results are now presented below.
>
> | Model_dag                              | Problems | Pass@turn1 | Syntax@turn1 | Pass@turn2 | Syntax@turn2 | Pass@turn3 | Syntax@turn3 |
> |------------------------------------|----------|------------|--------------|------------|--------------|------------|--------------|
> | DeepSeek-R1_dag                    | 100      | 1.00%      | 52.00%       | 4.00%      | 79.00%       | 6.00%      | 79.00%       |
> | DeepSeek-V3.1-Terminus_dag         | 100      | 1.00%      | 59.00%       | 3.00%      | 94.00%       | 7.00%      | 98.00%       |
> | DeepSeek-V3_dag                    | 100      | 0.00%      | 67.00%       | 4.00%      | 96.00%       | 4.00%      | 98.00%       |
> | QwQ-32B_dag                        | 100      | 0.00%      | 29.00%       | 0.00%      | 42.00%       | 2.00%      | 49.00%       |
> | Qwen2.5-Coder-32B-Instruct_dag     | 100      | 0.00%      | 51.00%       | 1.00%      | 62.00%       | 1.00%      | 63.00%       |
> | Qwen3-Coder-480B-A35B-Instruct_dag | 100      | 0.00%      | 69.00%       | 1.00%      | 95.00%       | 4.00%      | 97.00%       |
> | gpt-4.1_dag                        | 100      | 1.00%      | 26.00%       | 2.00%      | 77.00%       | 4.00%      | 96.00%       |
> | gpt-4o_dag                         | 100      | 0.00%      | 69.00%       | 0.00%      | 99.00%       | 2.00%      | 99.00%       |
> | o4-mini_dag                        | 100      | 0.00%      | 22.00%       | 1.00%      | 76.00%       | 4.00%      | 93.00%       |
>
> As shown, we observe a persistent **Syntax–Pass gap** across models and refinement turns: while **Syntax@turn3** ranges from **49–99%**, **Pass@turn3** remains only **1–7%**. Moreover, pass rates on **DAG** topologies are, on average, **lower** than on the chain-structured setting under the same refinement method. This reinforces that the failures of compositional reasoning are driven by difficulties in coordinating **interdependent contracts**.
>
> Thank you again for the suggestion and we will include this analysis in the main text, discussing its implications for scaling to more diverse topologies.

---

> ### Author Response · Authors · 2025-11-21
> **Weakness3**
>
> > **Weakness3:** There is a lack of evaluation on more advanced test-time scaling strategies, such as multi-turn self-refinement.
>
> We appreciate this suggestion and have conducted further experiments to evaluate an advanced test-time scaling strategy based on **multi-turn self-refinement**.
>
> Concretely, we set a refinement threshold of **3 turns**: if the model has not resolved the issue within 3 refinement steps, it outputs the final answer from the last turn. At each refinement step, we feed the model both the error message from the verifier and its previous output, asking it to improve its next response. We provide the refinement prompt and the corresponding quantitative results below.
>
> ```text
> "The Dafny program you produced failed verification. Here is your previous answer:\n"
> f"```dafny\n{previous_output}\n```\n"
> "Dafny verifier responded with:\n"
> f"```\n{previous_feedback}\n```\n"
> "Please fix the Dafny program so that it verifies, keeping the exact signature, requires, and ensures. "
> "Return only the Dafny program enclosed in ```dafny``` fences."
> ```
>
> | Model                          | Problems | Pass@turn1 | Syntax@turn1 | Pass@turn2 | Syntax@turn2 | Pass@turn3 | Syntax@turn3 |
> |--------------------------------|----------|------------|--------------|------------|--------------|------------|--------------|
> | DeepSeek-R1                    | 300      | 0.33%      | 85.67%       | 5.67%      | 94.33%       | 9.67%      | 95.00%       |
> | DeepSeek-V3.1                  | 300      | 0.00%      | 49.00%       | 1.67%      | 91.67%       | 5.33%      | 96.00%       |
> | DeepSeek-V3                    | 300      | 0.00%      | 74.33%       | 2.67%      | 95.00%       | 5.00%      | 96.00%       |
> | QwQ-32B                        | 300      | 0.00%      | 44.67%       | 0.00%      | 56.67%       | 1.00%      | 61.67%       |
> | Qwen2.5-Coder-32B-Instruct     | 300      | 0.33%      | 59.67%       | 0.33%      | 62.33%       | 0.67%      | 63.67%       |
> | Qwen3-Coder-480B-A35B-Instruct | 300      | 0.33%      | 78.33%       | 0.67%      | 93.33%       | 3.00%      | 95.67%       |
> | gpt-4.1                        | 300      | 0.00%      | 46.67%       | 1.33%      | 86.67%       | 3.33%      | 92.67%       |
> | gpt-4o                         | 300      | 0.33%      | 94.00%       | 2.00%      | 98.33%       | 4.67%      | 98.67%       |
> | o4-mini                        | 300      | 0.00%      | 37.33%       | 2.00%      | 92.00%       | 9.67%      | 96.00%       |
>
> As shown, we find the multi-turn self-refinement method to be helpful. Although its improvement on the **Syntax Rate** is limited, the **Pass Rate** generally increases once the verifier provides accurate feedback on the errors. At the same time, success rates remain far from saturation, indicating that the benchmark **remains challenging** even with verifier feedback.
>
> Thank you again for the suggestion and we will include the additional evaluation results in the main text.

---

> ### Author Response · Authors · 2025-11-21
> **Weakness4**
>
> > **Weakness4:** The evaluation described in the main text does not consistently standardize or control details about compute and prompting across models (tokens, temperature, verifier retry policy).
>
> We agree that standardizing compute details is important for a fair comparison, and we have already updated the specific settings in the **Appendix H (”Compute Settings”)**, and revised the main text to state “*We employ a unified prompt together with standardized compute settings across all evaluations (see Appendix H and Appendix G)”*. On the other hand, we want to clarify that the **prompts** used during evaluation were fully standardized across all models, as detailed in **Appendix G (“Prompt for Evaluation”)**.
>
> Concretely, we enumerate the details about **token budgets**, **inference sampling (including temperature)**, **verifier retry policy**:
>
> **Token budgets:** Many of the evaluated systems are reasoning-oriented models, which makes strict control over the number of generated tokens inherently difficult: different models follow different reasoning trajectories, and models like R1 in particular tend to iterate, backtrack, and reconsider their answers on this type of task. As a result, token usage is largely emergent rather than something we can tightly equalize across models. For illustration, the table below reports the token counts for several representative individual R1 outputs. Given this variability, we did not impose any additional hard constraint on the number of generated tokens when calling the APIs. Instead, for each model we set the **max_output_tokens** (or equivalent) parameter to the **largest value** allowed by the corresponding provider, so that the model’s reasoning process would not be artificially truncated by our configuration.
>
> | Model                       | InputTokens | OutputTokens |
> |-----------------------------|------------:|-------------:|
> | Pro/deepseek-ai/DeepSeek-R1 | 2509        | 19564        |
> | Pro/deepseek-ai/DeepSeek-R1 | 2509        | 16746        |
> | Pro/deepseek-ai/DeepSeek-R1 | 2465        | 17918        |
> | Pro/deepseek-ai/DeepSeek-R1 | 2330        | 17507        |
> | Pro/deepseek-ai/DeepSeek-R1 | 2493        | 15952        |
> | Pro/deepseek-ai/DeepSeek-R1 | 2509        | 22216        |
> | Pro/deepseek-ai/DeepSeek-R1 | 2544        | 13654        |
> | …                           | …           | …            |
>
>
> **Inference sampling settings:** We set the default decoding parameters to **a temperature of** **0.7** and **a top-p of** **0.8** for all models; for **Claude models**, whose API does not expose **top-p**, we used the provider’s default, and for **o4-mini**, which only supports a **fixed temperature of** **1** and has **no top-p parameter**, we used its default configuration. These moderate, near-default settings are commonly used in practice and provide a reasonable trade-off between stability and diversity while keeping the settings as comparable as possible across models.
>
> **Verifier Retry Policy:** The verifier retry policy in the current setup allows exactly **one verification attempt per output**, with a **fixed timeout threshold of 60s**. If the SMT solver does not finish within 60s, we consider the instance impractical to solve in our setting and do not count it as a success.

---

### Author Response · Authors · 2025-12-02
**Comment Summary for AC**

**Dear Area Chair,**

We post this author comment to succinctly summarize our updates to support your meta-review.

**Recognition of contributions.** Reviewers broadly agree that the work offers a timely, realistic diagnostic of **compositional** formal verification in multi-function settings: it fills the gap beyond single-function benchmarks (**Reviewer** **KdjF**), provides clear ground truth with quantitative evidence of the syntax–verification gap (**Reviewer M3WA**), includes broad evaluation with an insightful failure taxonomy (**Reviewer mpGN**), and directly probes compositional reasoning where current models struggle (**Reviewer 2F9W**).

**Key rebuttal updates.**

1. **Scalability & Topology breadth.** The pipeline is **template-based**, it scales to new topologies and data pools and ports to **Verus/Boogie** via front-end/template swaps. As a concrete instantiation, we extend the benchmark with **100 DAG-structured programs** (10 non-chain templates), and **this extension shows** a persistent **Syntax–Pass gap** with **lower Pass rates** than in **chain-structured** settings, indicating that the core difficulty lies in coordinating **interdependent contracts** rather than formatting.
2. **Test-time scaling baseline.** **We add** a multi-turn verifier-in-the-loop self-refinement baseline; Pass rates improve but remain far from saturation, indicating that DafnyComp is challenging **even with more advanced test-time scaling strategies**.
3. **Fairness & standardization.** **We include** the model-selection record in **appendix** and, **in the newly added experiments, we** **exclude Claude-family results** to avoid same-family generation/evaluation bias. We also clarify our **standardized compute protocol** for all models: one unified evaluation prompt, a uniform decoding policy (falling back to provider defaults only when a parameter isn’t exposed), a fixed verification procedure (**one** verifier run per output under a fixed timeout), and a high output budget to avoid truncation (we intentionally do **not** normalize token lengths across models).”
4. **Robustness checks.** **We add** two checks: (i) **flattening** each composition to an **equivalent single-function** leaves verified success essentially unchanged; (ii) **longer solver timeouts** do not alter outcomes. Together, these indicate the results are not artifacts of style choices or prover limits.

**Note on Reviewer 2F9W’s concerns.** These points are **covered in the “Key rebuttal updates”**. Flattening each composition to a single-function form and re-running timed-out cases with longer budgets yield unchanged verification outcomes, indicating our conclusion is not driven by composition form or SMT limits. In addition, we **clarify test construction** (by intersecting constituent I/O constraints), and we have **revised wording, captions, and examples** (appendix context, “not a typo,” and the 300 vs. 900 clarification). All edits are marked **in blue** in the updated version.

We hope these added experiments, clarifications, and manuscript updates clarify the contribution and evidence and address the key concerns. We thank the reviewers for constructive feedback that improved the paper’s clarity and presentation, and we appreciate your time and consideration.

Best wishes,

Authors of Submission19018

---

### Meta-Review · Area_Chair_3LZC · 2026-01-07

**Summary:**

The paper introduces a new benchmark, DAFNYCOMP, which targets compositional specification generation in Dafny using multi-function programs. Reviewers recognize that the benchmark focuses on a novel compositional setting and that the paper provides comprehensive case studies with insightful error analysis.

Reviewers 1 and 3 raise concerns about the limited program topologies, noting that the benchmark focuses on chain structures. Reviewer 3 further questions how chained multi-function programs differ from single-function settings. Reviewers 1 and 4 are concerned about the lack of evaluation of more advanced test-time scaling strategies. Reviewer 3 also notes that SMT solvers may fail even when specifications are correct.

The authors’ rebuttal provides substantial new evidence, including evaluation results of multi-turn refinement methods, new non-chain DAG-structured programs, and single-function equivalence. These additions largely address the major concerns raised in the reviews.

I have a minor remaining concern about potentially overstating the extent to which performance drops are due to compositionality, since generating specifications from code alone is already challenging. But I am OK with a new specification benchmark with less/without contamination.

Given the high likelihood that reviewers will raise their scores, the overall assessment is likely to be positive, and I lean toward acceptance.

**Reviewer Concerns:**

Concerns about the possibility that the Dafny prover may fail even when the verification condition is correct are not addressed; increasing the timeout will not turn such failure cases into successful ones.

Other concerns are largely addressed.

**Reviewer Scores:**

Reviewers 1, 3, and 4 are likely to raise their scores, as their reviews indicate openness to discussion and the rebuttal appears to address their main concerns.

---

### Decision · Program_Chairs · 2026-01-26

Accept (Poster)